# Sublimation Measurements of Tundra and Taiga Snowpack in Alaska

Kelsey A. Stockert[1], Eugénie S. Euskirchen[2], Svetlana L. Stuefer[1]

[1]Department of Civil, Geological and Environmental Engineering, Water and Environmental Research Center, College of Engineering and Mines, University of Alaska Fairbanks, Fairbanks, AK 99775, USA

[2]Institute of Arctic Biology and Department of Biology and Wildlife, University of Alaska Fairbanks, Fairbanks, AK 99775, USA

*Correspondence to*: Svetlana Stuefer (sveta.stuefer@alaska.edu)

**Abstract.** Snow sublimation plays a fundamental role in the winter water balance. To date, few studies have quantified sublimation in tundra and boreal forest snow by direct measurements. Continuous latent heat data collected with eddy covariance (EC) measurements from 2010 to 2021 were used to calculate snow sublimation at six locations in northern Alaska: three Arctic tundra sites at distinct topographical and vegetation communities in the Imnavait Creek watershed on the North Slope underlain by continuous permafrost, and three lowland boreal forest/taiga sites in discontinuous permafrost in interior Alaska near Fairbanks. Mean surface sublimation rates range from 0.08–0.15 mm day$^{-1}$ and 15–27 mm year$^{-1}$ at the six sites, representing, on average, 21% of the measured solid precipitation and 8–16% of the cumulative annual water vapor flux to the atmosphere (evaporation plus sublimation). The mean daily sublimation rates of the lowland boreal forest sites are higher than those of the tundra sites, but the longer snow cover period of the tundra sites leads to greater mean annual sublimation rates. We examined the potential controls, drivers, and trends of the sublimation rates by using meteorological data collected in conjunction with EC measurements. This research improves our understanding of how site conditions affect sublimation rates and highlights the fact that sublimation is a substantial component of the winter hydrologic cycle. In addition, the study contributes to the sparse literature on tundra and boreal sublimation measurements and the measured rates are comparable to sublimation estimates in other northern climates.

## 1 Introduction

Snow sublimation is the phase change from snow grains in the snowpack to water vapor in the atmosphere (Fierz et al., 2009). It is a fundamental process in the winter water balance that affects the amount of snow on the ground at the end of winter (Bowling et al., 2004; Molotch et al., 2007; Pomeroy & Essery, 1999; Reba et al., 2012). Studies estimate that sublimation is responsible for between 0.1% and 90% of snow mass loss to the atmosphere (Stigter et al., 2018). Liston and Sturm (2004) estimate that 10–50% of annual snowfall in the Arctic sublimates. These large sublimation variations are due to the local and regional differences in environmental conditions that control snow sublimation (e.g., air temperature, wind speed, humidity, and solar radiation).

Sublimation is important because it affects the amount of seasonal snow that accumulates on the ground during winter periods. In northern Alaska, snow can be present on Earth's surface for most of the year. Snow affects permafrost, thermal properties, and freezing rates of lakes and sea ice, soil microbiology, soil chemistry, the animals that spend winter under the snow, humans, and infrastructure (Gray & Male, 1981). Snow affects how much solar radiation is absorbed by Earth and how much is reflected, a fundamental process in the global climate and a key component of global warming (Loaiciga et al., 1996). Snow also affects water supply and how much water is available for human activities and resources. A recent example from 2021 highlights the importance of understanding the relationship between snow, sublimation, groundwater, and streamflow: the Colorado River snowpack was estimated at 90% of average, but streamflows were only 36% of average. It is currently speculated that the discrepancy may in part be explained by sublimation (Lundquist et al., 2024).

Understanding the factors controlling sublimation in different climatic regions and snow classes will improve our understanding of how site conditions affect sublimation rates. Global seasonal snow classes (tundra, boreal forest, montane forest, maritime, prairie, and ephemeral) are used to put local Alaska study sites into a global seasonal snow perspective (Sturm & Liston, 2021). Globally, almost half of Earth's terrestrial area is covered by tundra and boreal forest snow classes, at 31.8% and 28.3%, respectively (Sturm & Liston, 2021). Both blowing snow and static sublimation are common winter processes in tundra environments (Liston & Sturm, 2004), while canopy snow sublimation is a characteristic feature of forested environments (Pomeroy et al., 1998).

The winter water balance (in the absence of wind transport) is simple: snow water equivalent (SWE) equals precipitation minus sublimation (Liston & Sturm, 2004; Stuefer et al., 2020). But, in the Arctic, using field observations to make this moisture budget calculation produces sublimation estimates that are wide-ranging and unreliable due to errors associated with solid precipitation measurements; systematic biases in solid precipitation measurements include wind undercatch, wetting loss, and evaporation loss (Fassnacht 2004; Goodison et al. 1998; Nitu et al. 2018). Sublimation can also be estimated by solving energy balance equations; including the Penman Monteith, bulk aerodynamic, and aerodynamic profile methods (Marks et al., 2008; Sexstone et al., 2016; Stigter et al., 2018); or direct measurements.

Direct sublimation measurements can be in the form of a sublimation pan, a snow pillow, structure-in-motion photogrammetry, or the eddy covariance (EC) method. Sublimation pans require manual measurements and are not feasible for long-term studies in remote (unattended) locations (Guo et al. 2018; Herrero & Polo, 2016). Snow pillows pose numerous problems that reduce the sublimation measurement accuracy: they alter snow conditions from the surrounding area and create snow bridges, they do not work well in shallow snowpacks, and they are adversely affected by high wind speeds (Herrero & Polo, 2016). There are recent advances in estimating sublimation rates through measurements of snow depth and volume change using time-lapse structure-from-motion photogrammetry methods (Liu et al. 2024), but this method cannot quantify blowing snow or canopy sublimation rates. EC measurements are the most direct means available to measure vertical turbulent fluxes (Marks et al., 2008; Molotch et al., 2007; Reba et al., 2009, 2012; Sexstone et al., 2016; Stigter et al., 2018). However, EC towers that operate year-round are rare in much of Alaska due to challenges associated with the complexity and expense of maintenance during the harsh winter. The study presented herein uses EC tower data to analyse northern Alaska snowpack sublimation established in locations representative of the tundra and boreal forest snow classes (Figure 1).

To the authors' knowledge, there are no published studies that have calculated sublimation using the EC method for durations greater than 3 years anywhere in the world. In this study, twelve years of EC measurements from six sites in northern Alaska, distinguished by snow classes, vegetation communities, and permafrost (Figure 1), were analysed to 1) quantify the magnitude of snow sublimation, 2) assess spatial and temporal variability, 3) compare sublimation rates with other water fluxes, and 4) investigate drivers of sublimation using meteorological and environmental data.

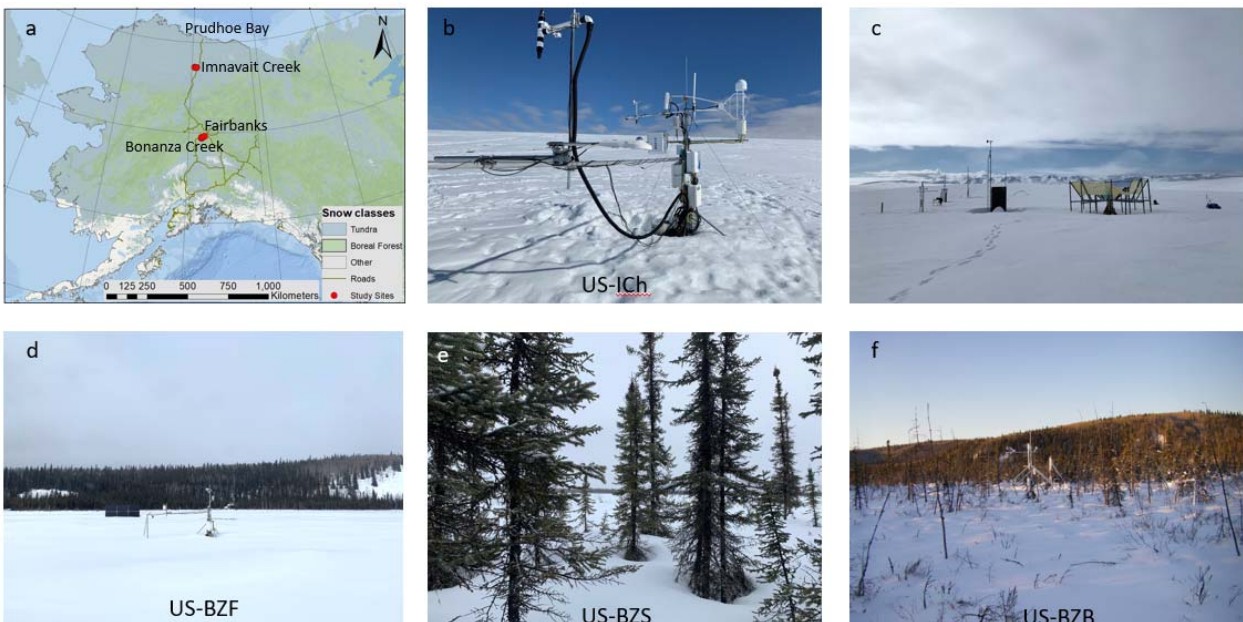

**Figure 1: Location of study sites within the Sturm and Liston (2021) tundra and boreal forest snow classes in northern Alaska (a); Arctic tundra EC tower at Imnavait Creek (b); SNOTEL and UAF weather stations at Imnavait Creek (c); snow cover at a boreal fen (US-BZF) EC tower in the Alaska Peatland Experiment (APEX), which is associated with the Bonanza Creek Long Term Ecological Research program (d); boreal forest black spruce site (US-BZS) at APEX (e); boreal forest thermokarst bog site (US-BZB) at APEX (f).**

## 2 Background

### 2.1 Study Area

This study used data from six sites in northern Alaska: three Arctic tundra sites on the North Slope in the northern foothills of the Brooks Range and three lowland boreal forest sites in the subarctic Interior (Figure 2). The sites are referenced in this paper

by their AmeriFlux site ID (https://ameriflux.lbl.gov/). AmeriFlux is a network of EC research sites across the Americas. Monthly summaries of wind speed, air temperature, and precipitation in Table 1 compare the meteorological settings of the tundra and boreal forest environments. In brief, there are comparable precipitation normals between the two regions, but the tundra climate is substantially windier than the lowland boreal forest. Mean daily air temperatures remain below freezing from October to April in tundra sites and from October to March in lowland boreal forest sites (Table 1).

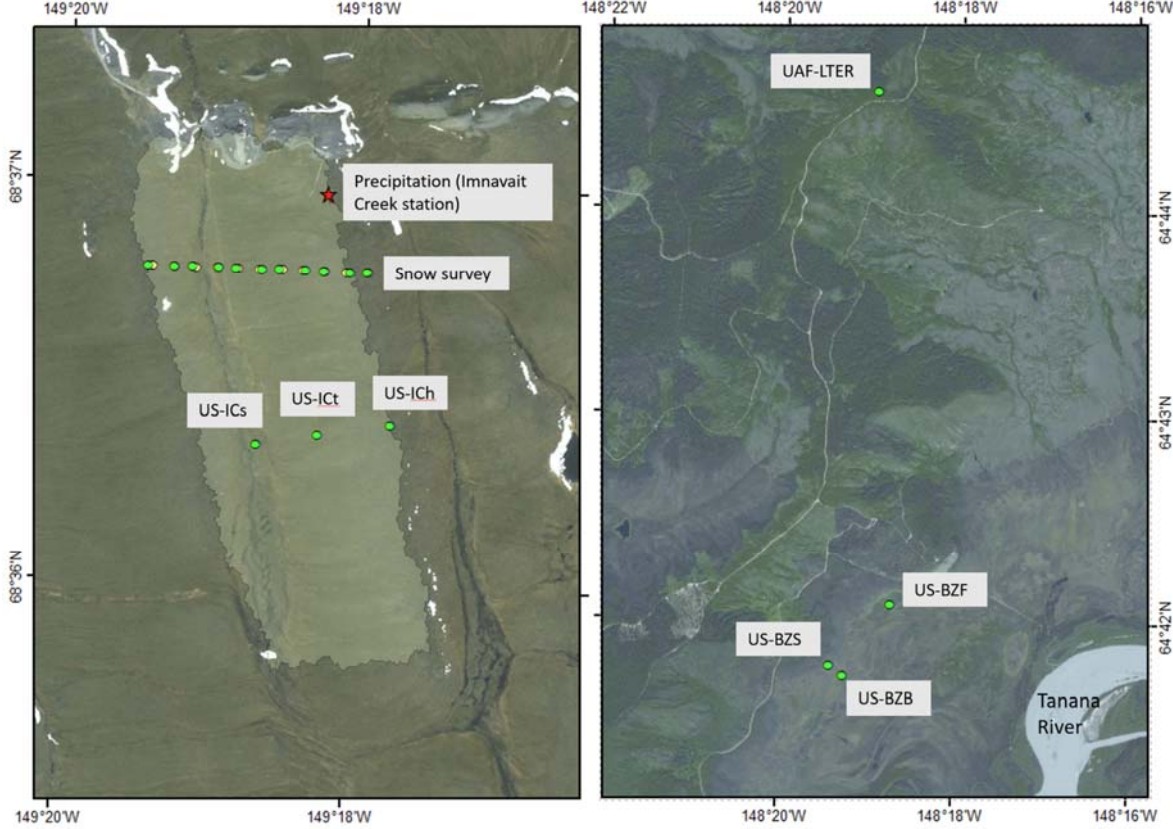

**Figure 2: Location of study sites: tundra sites at Imnavait Creek watershed (a); lowland boreal forest sites at the Alaska Peatland Experiment, part of the Bonanza Creek Long Term Ecological Research program (b). Basemaps credited SPOT 5 Image Corporation.**

**Table 1: Monthly meteorological summaries for Arctic tundra and lowland boreal forest regions, as measured from the Eddy Covariance towers. Mean air temperature, mean wind speed, and max wind speed are averages of the three sites in each region.**

|  | Mean Daily Air Temperature (°C) |  | Mean Daily Wind Speed (m s⁻¹) |  | Max Daily Wind Speed (m s⁻¹) |  | Total Precipitation Normal[1] (mm) |  |
|---|---|---|---|---|---|---|---|---|
|  | Tundra | Lowland Boreal | Tundra | Lowland Boreal | Tundra | Lowland Boreal | Tundra | Lowland Boreal |
| January | -18 | -20 | 2.6 | 0.9 | 22.7 | 12.6 | 9 | 15 |
| February | -17 | -16 | 2.9 | 1.1 | 20.1 | 8.3 | 13 | 13 |
| March | -16 | -10 | 2.6 | 1.3 | 15.9 | 8.8 | 9 | 10 |
| April | -9 | 1 | 2.5 | 1.5 | 12.7 | 8.5 | 10 | 9 |
| May | 0 | 11 | 2.4 | 1.5 | 14.5 | 7.6 | 18 | 14 |
| June | 8 | 16 | 2.5 | 1.4 | 12.2 | 6.2 | 46 | 38 |
| July | 10 | 17 | 2.4 | 1.3 | 26.9 | 21.4 | 80 | 57 |
| August | 6 | 13 | 2.3 | 1.2 | 11.2 | 6.3 | 72 | 53 |
| September | 0 | 7 | 2.3 | 1.1 | 14.4 | 8.8 | 33 | 34 |
| October | -7 | -1 | 2.3 | 1.0 | 12.9 | 13.0 | 23 | 19 |
| November | -15 | -12 | 2.6 | 1.0 | 19.5 | 18.4 | 14 | 19 |
| December | -18 | -16 | 2.4 | 0.9 | 15.9 | 10.8 | 12 | 14 |
| **Annual** | **-6.3** | **-0.8** | **2.5** | **1.2** | **16.6** | **10.9** | **339** | **295** |

1. Fairbanks Station USW0002641 and Imnavait Creek Station USS0049T01S, 1991–2020, https://akclimate.org/data/precipitation-normals/

### 2.1.1 Arctic Tundra Sites at Imnavait Creek Watershed

A network of three eddy flux towers was established in the Imnavait Creek watershed in 2011 to measure carbon, water, and energy fluxes along a hillslope moisture gradient (Euskirchen et al., 2017). The Imnavait Creek watershed, a small arctic

watershed (2.2 km$^2$), is located in the foothills of the Brooks Range at 68°37′N, 149°18′W, and 770–980 m above sea level (Figure 2a). Within the watershed, the towers are located on a gently rolling hill less than 0.5 km from each other along a topographic sequence from valley bottom to ridge and within distinct vegetative communities: wet sedge (Us-ICs), tussock (US-ICt), and dry heath (US-ICh), respectively (Euskirchen et al., 2017; Walker et al., 1994).

The landscape is treeless with rolling hills, broad valleys, and continuous permafrost. Imnavait Creek is a small, beaded tributary of the Kuparuk River. The mean annual air temperature (MAAT) during the study period is –6.3°C and mean annual precipitation (MAP) is 339 mm, with 40% of that occurring as snow (Table 1). Mean monthly air temperatures are below freezing from September/October to May, and generally, snowpack is present during those same months (Stuefer et al., 2020).

Snow cover in the Imnavait Creek area is representative of tundra snow class; that is, windblown with drifts, hard packed, cold, dry, and thin (Sturm & Liston, 2021; Brown et al., 2021). The snow covers low stature vegetation (< 0.5 m) in the treeless, exposed, windy environment. Large spatial variability in snow depth is a common feature of the tundra snow class (Benson & Sturm, 1993).

Predominant winds in the Imnavait Creek watershed are from the south and west and can exceed 20 m s$^{-1}$ (Sturm & Stuefer, 2013; Table 1), creating deep, dense drifts in depressions on the lee side of landscape features (Parr et al., 2020). The snowpack structure often consists of low-density depth hoar at the base, covered by a hard wind slab layer on top (Benson & Sturm, 1993). At the end of winter in late April from 1985–2017, mean snow depth at the Imnavait Creek watershed was 50 cm and average SWE was 125 mm (Stuefer et al., 2020). Snow and wind conditions at this watershed are similar to those throughout the gently rolling foothills of the northern Brooks Range (Sturm & Stuefer, 2013).

### 2.1.2 Lowland Boreal Forest Sites at the Alaska Peatland Experiment

The lowland boreal forest sites are in the Tanana Flats of interior Alaska, approximately 30 km southeast of Fairbanks at 64°42′N, 148°19′W (Figure 2b). These sites are associated with the Bonanza Creek Long Term Ecological Research Program (lter.uaf.edu) and are part of the Alaska Peatland Experiment (APEX), which began in 2005 as an effort to understand water and carbon cycling in a rich fen (Turetsky et al., 2008) and has expanded to include thermokarst bogs and black spruce peat plateau areas (Euskirchen et al., 2014). As with the tundra sites, the boreal forest sites are in close proximity at 0.5 km apart, in distinct ecosystems and permafrost regimes.

With trees ~100 years old, the US-BZS site (Figure 2b) is in a mature black spruce forest (*Picea mariana*) that overlays an intact peat plateau of cold soils that rises ~130 cm from the surrounding landscape. US-BZF is a rich fen composed of grasses, sedges, and forbs, but lacks trees and permafrost. The US-BZB site is in a collapsed scar bog within a circular depression that formed through thermokarst-related processes (subsidence resulting from ground ice thaw). The site contains active thaw margins with significant dieback of the black spruce.

Interior Alaska has a subarctic continental climate. Typically, snowpack is present from mid–late October through mid–late April. The study period MAAT was –0.8°C and the MAP was 295 mm, as measured at the Fairbanks International Airport (Geophysical Institute - University of Alaska Fairbanks, 2023), with 45% of the precipitation occurring as snow from October through April (Table 1).

These sites are representative of the boreal forest snow class (Sturm & Liston, 2021). Sometimes called taiga snow, the snow here is characterized as thin, dry, and low density, consisting mainly of depth hoar by the end of winter (Sturm & Benson, 1997). This snow class is found in forested environments, where there is less wind action than on the tundra snow (Sturm & Liston, 2021; Table 1). Wind is typically low in the Tanana Flats, with strong inversions present.

**2.2 Types of Sublimation**

Total sublimation equals static-surface sublimation plus blowing-snow sublimation plus canopy-interception sublimation (Molotch et al., 2007). During wind transport, blowing snow particles sublimate, but EC primarily measures only the turbulent fluxes of static-surface sublimation and does not directly measure blowing snow sublimation (Lackner et al., 2022; Reba et al., 2012; Stigter et al., 2018). This possible underestimate will be examined further in the Discussion section. Canopy sublimation takes place where snow is captured in tree canopies, but five of the six EC sites in this study are in low-growing vegetation environments where plants are completely covered by snow during the winter season so that the canopy sublimation term does not apply.

**3 Data and Methods**

**3.1 Eddy Covariance Sublimation Processing and Calculations**

The EC technique measures turbulent fluxes between the land and atmosphere to calculate fluxes of gases, water, and heat per unit time (Burba & Anderson, 2008). EC towers at each of the six sites are equipped with a 3-D sonic anemometer and an infrared gas analyser (IRGA) that measure the latent heat fluxes 10 times per second (10 Hz) 2.5–5 m above the ground (and above the canopy). The instrument configurations, set-up, and data processing at the tundra and boreal forest sites have been fully described in Euskirchen et al. (2012, 2014, 2017, 2020, 2024) and are available through the AmeriFlux database. Sublimation calculations use both filtered latent heat measurements (70%) and gap-filled data (30%). Filtering primarily refers to removing data when there is optical impedance by precipitation or aerial contaminants. This is denoted by the automatic gain control (AGC) values measured by the infrared gas analysers. These values are used as a quality assurance/quality control variable for both flux and radiation data, with 60% as the maximum threshold AGC value. Data gaps occur from instrument malfunction, instrument calibration, or occasional power outages in winter months. For data gaps of 1–6 days, missing observations were replaced by the mean for that time period (half hour) and based on adjacent days using the ReddyProc software (Euskirchen et al. 2024). For data gaps of 1–2 weeks, marginal distribution sampling is used to fill missing data (Euskirchen et al., 2024). When compared, mean daily sublimation rates with only the filtered data were identical to within one hundredth of a millimetre to the gap-filled data. Due to prolonged power outages and equipment malfunction, some water years (defined as October 1–September 30) have missing data that were unable to be gap filled. Water years with missing data were not included in our analyses. Complete water years are listed in Table 2 in the Results. This is a unique dataset; there are few long-term EC systems operating year-round in northern regions, particularly in the Arctic tundra.

Measured latent heat flux can be converted to half-hour averages of water vapor flux (mm) by dividing latent heat by either the latent heat of sublimation to derive sublimation and deposition ($2.838 \times 10^6$ J kg$^{-1}$) or the latent heat of vaporization for evapotranspiration (ET) and condensation ($2.454 \times 10^6$ J kg$^{-1}$). Sublimation and ET are calculated when the flux is positive (meaning direction of flux is to the atmosphere), while condensation and deposition are calculated when the flux is negative. Sublimation and deposition are calculated when snowpack is present; snowpack presence and snow cover duration are determined from the albedometer installed on the EC towers and from webcam images at the sites. Annual sublimation for each year is calculated based on days with snow cover present during each snow season. Hourly, daily, monthly, and annual sublimation rates are cumulative values that represent the sum of half-hour sublimation rates over a corresponding time period: hour (mm hour$^{-1}$), day (mm day$^{-1}$), month (mm month$^{-1}$), and year (mm year$^{-1}$).

**3.2 Meteorological and Snow Data**

Meteorological data are collected at each EC tower at 15-second intervals and averaged over 30-minute periods. This study utilized air temperature, soil temperature, relative humidity, net radiation, albedo, wind speed, temperature gradient, and vapor

pressure deficit (VPD) for processing and analysis. Temperature gradient is the soil temperature minus air temperature, or the gradient through the snowpack between the ground surface and a sensor approximately 2–5 m above ground surface (depending on the site). VPD is the difference between saturated vapor pressure and the actual vapor pressure and is calculated automatically at the stations; a higher VPD indicates drier atmospheric conditions.

Snow data used in this study include solid precipitation, SWE, and snow cover duration. At Imnavait Creek (Figure 2a), the U.S. Department of Agriculture (USDA), Natural Resources Conservation Service (NRCS), measures solid precipitation at SNOTEL site number 968. This storage gauge has a Wyoming wind shield (Figure 1c) and reports automated daily precipitation measurements (https://wcc.sc.egov.usda.gov/nwcc/site?sitenum=968). Also at Imnavait Creek, UAF measures end-of-winter SWE accumulation by collecting 900 snow depth and 50 snow density measurements across the watershed (Stuefer et al., 2020). These observations are made along the same snow course every year, and are used to calculate watershed average SWE (Figure 2a). These snow depth, snow density, and SWE data are available at the Arctic Data Center (Stuefer et al., 2019).

At the APEX sites (Figure 2b), SWE is measured with an automated snow pillow managed by the UAF Long Term Ecological Research (LTER) program (https://www.lter.uaf.edu/data/site-detail/id/52). Solid precipitation is not measured at APEX sites; therefore, we used precipitation measurements from the NOAA, NWS (National Oceanic and Atmospheric Administration, National Weather Service) weather station USW0002641 at Fairbanks International Airport at 64°48′N, 147°52′W (https://www.ncdc.noaa.gov/cdo-web/datasets/GHCND/stations/GHCND:USW00026411/detail).

**3.3 Statistical Methods**

Standard statistical methods are applied to analyse relationships, trends, and differences in sublimation rates between sites and snow classes (Gottelli & Ellison, 2004).

1. The magnitude of daily, monthly, and annual sublimation rates are calculated for water years with complete records. Mean values, standard deviation, and standard error of the mean are used to compare the variability in sublimation rates between sites (Section 4.1), to compare sublimation with environmental conditions (snow cover duration, SWE, solid precipitation) (Section 4.2.1), and to evaluate sublimation rates with other water fluxes (ET, condensation, and deposition) (Section 4.2.2).

2. A one-way Welch's Analysis of Variance (ANOVA) tests are performed to determine whether significant differences exist in annual sublimation rates across the six individual sites (Section 4.3). Because analyses are restricted to years of complete data (more details in Section 3.1 and listed in Table 2), there are unequal numbers of annual sublimation rates at the six sites. The Welch's ANOVA is a more robust test than a traditional ANOVA since it does not assume equal variances and works well with unequal group sizes to ensure valid comparisons between the sites' mean sublimation rates. Post hoc Games-Howel tests are used to identify pairwise differences between group means. Data are log transformed to normalize the positively skew.

3. A Welch's t-test are used to test differences in annual sublimation rates between two snow classes (Arctic tundra and boreal forest) and between sites with and without a canopy (Section 4.3). As with Welch's ANOVA, this method accounts for unequal variances. Data are log transformed to normalize the positively skew.

4. Pearson's correlation coefficient (r), single ordinary least squares (OLS), and multiple linear regressions (MLR) with forward selection model are applied to evaluate the relationship between hourly and daily sublimation rates and meteorological and environmental variables (Section 4.4). The meteorological variables are added in the stepwise regression in the following order for the MLR: 1) air temperature and wind speed, 2) vapor pressure deficit (VPD) and wind speed, 3) air temperature, wind speed, and relative humidity, 4) air temperature, VPD,

net radiation, temperature gradient, and wind speed. Regression models are evaluated based on their p-values, $r^2$, and adjusted $r^2$. All statistical methods use a significance level of 0.05.

As noted above, data are summarized at different time scales: hourly, daily, monthly, and annual. Reporting daily rates is valuable for comparison with findings from other studies in the literature (Section 5.3). Correlations and regressions with meteorological variables are conducted using the hourly and daily data because these relationships are stronger at finer temporal resolutions. In contrast, regressions with environmental variables, namely snow cover duration, SWE, and solid precipitation, are meaningful (and available) only at the annual scale. ANOVA tests are applied to annual data to provide a clearer understanding of the impacts of sublimation over entire winters; and avoids the limitations of daily rates, which fail to capture the substantial difference in the snow class's snow cover duration (see specifics of differences in Section 4.2.1).

Lastly, most analyses group the data by snow class. This approach reflects the greater influence of snow-climate conditions over individual site meteorological conditions on sublimation rates. Sites within the same snow class are located only 0.5 km apart with similar weather conditions, whereas the snow classes are over 600 km and represent distinct climate zones (Shulski & Wendler, 2007).

## 4 Results

### 4.1 Daily, Monthly, and Annual Sublimation Rates

For the twelve-year period, mean daily sublimation rates are 0.08–0.10 mm day$^{-1}$ in tundra and 0.08–0.15 mm day$^{-1}$ in lowland boreal forest (Table 2). The daily sublimation variability is high for all sites: standard deviation values are greater than the mean, and mean daily rates are 5–7% of the maximum daily rate. Across the six sites, mean daily rates are highest at the black spruce site (US-BZS) in the boreal forest snow class (Table 2). Max daily sublimation rates are highest in the tundra sites (US-ICs and US-Ich), followed by boreal forest site (US-BZS).

**Table 2: Mean and maximum daily and annual sublimation rates in boreal forest and tundra snow.**

| Snow Sublimation Rates | | | | | | | |
|---|---|---|---|---|---|---|---|
| | | | | Daily (mm day$^{-1}$) | | Annual (mm year$^{-1}$) | |
| AmeriFlux Site ID | Description | Water Years with Complete Record | # Days with Sublimation Data | Mean ± SD | Max | Mean ± SD | Max |
| US-ICt | Tussock Tundra | 2013–2014, 2016–2021 | 1,761 | 0.10 ± 0.18 | 1.78 | 26 ± 7 | 38 |
| US-ICh | Dry Heath Tundra | 2010–2021 | 3,040 | 0.08 ± 0.13 | 2.25 | 20 ± 9 | 39 |
| US-ICs | Wet Sedge Tundra | 2010–2012, 2015–2021 | 2,518 | 0.10 ± 0.16 | 2.44 | 25 ± 12 | 49 |
| US-BZF | Rich Fen | 2015, 2017–2021 | 1,042 | 0.10 ± 0.16 | 1.92 | 17 ± 5 | 22 |
| US-BZB | Thermokarst Bog | 2014-2021 | 1,385 | 0.08 ± 0.13 | 1.52 | 15 ± 5 | 21 |
| US-BZS | Mature Black Spruce | 2011–2012, 2014–2021 | 1,106 | 0.15 ± 0.18 | 2.08 | 27 ± 6 | 35 |

Mean monthly (Figure 3) sublimation rates illustrate the variability over the snow season, over the duration of the study period, and between sites within tundra and boreal forest snow classes. There is a steady loss of water vapor to the atmosphere over the course of the winter from fall until spring. During the snow season, a range of 1.5–2.4 mm month$^{-1}$ of SWE sublimates at the tundra sites (Figure 3a) and 1.2–3.7 mm month$^{-1}$ range of SWE sublimates at the lowland boreal sites (Figure 3b). In the spring, prior to snowmelt, mean monthly sublimation increases to 5.4 mm month$^{-1}$ in May at tundra sites (Figure 3a) and to 7.2 mm month$^{-1}$ in April at lowland boreal sites (Figure 3b).

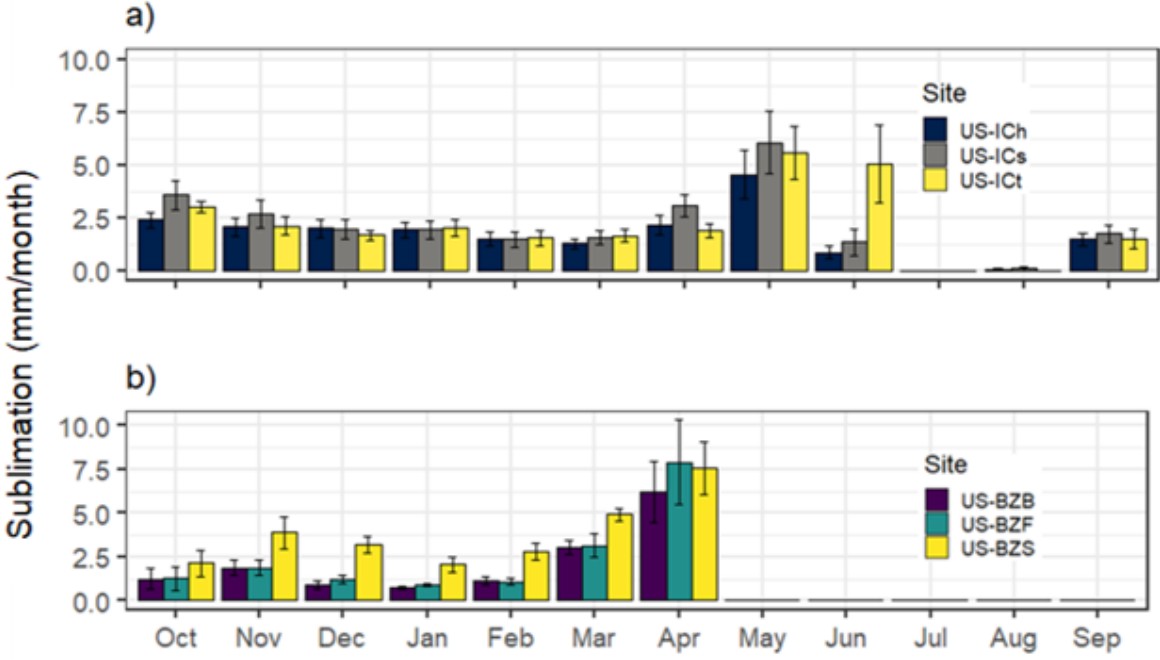

**Figure 3: Mean monthly sublimation throughout the water year at tundra sites (a) and lowland boreal forest sites (b). Error bars represent the standard error of the mean.**

Annually, mean sublimation rates are 20–26 mm year$^{-1}$ in tundra and 15–27 mm year$^{-1}$ in boreal forest (Table 2 and Figure 4). Annual standard deviation ranges from 22% to nearly 50% of the mean. Broadly, tundra site sublimation rates increase from

US-ICh (ridge) to US-ICs (valley bottom) to US-ICt (mid-slope). At the lowland boreal sites, sublimation rates are greatest at US-BZS (black spruce) and lowest at US-BZB (bog). There is high interannual variability in sublimation rates (Figure 4). The most extreme range at US-ICs measures a nearly 40 mm year$^{-1}$ difference of sublimated water between 2015 and 2021. The relative inter-site variability in sublimation rates within tundra and lowland boreal forest snow classes is lower than interannual variability (Figure 4).

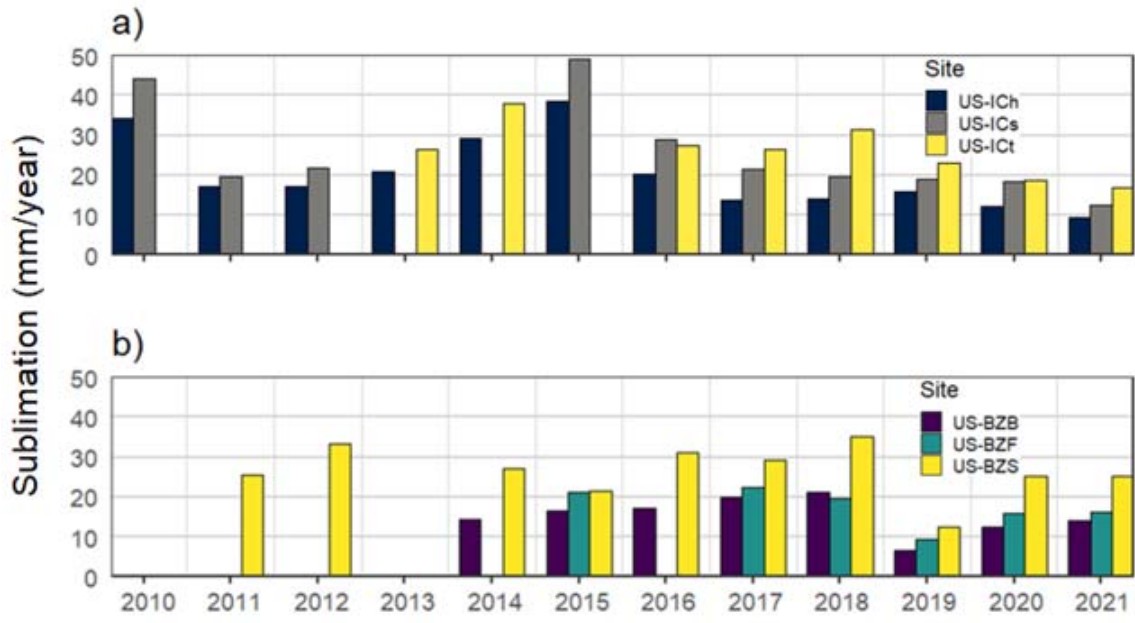

**Figure 4: Annual sublimation by water year at tundra sites (a) and lowland boreal sites (b).**

**4.2 Comparison of Annual Sublimation Rates with Other Water Fluxes**

**4.2.1 Snow Cover Duration, Snow Water Equivalent, and Solid Precipitation**

Snow cover duration at the tundra sites is approximately two months longer than at the lowland boreal sites. On average, snow cover duration is 254 days at tundra sites (mean date of snow onset is September 19 and snow melt is June 1) and 185 days at boreal sites (mean date of snow onset is October 19 and snow melt is April 22; Table 3). While the lowland boreal sites have comparable or higher mean daily sublimation rates than the tundra sites (Table 2), the longer snow cover period on the tundra means more days of sublimation and higher annual sublimation rates.

Overall, there is approximately two cm of SWE that sublimates throughout the winter in both tundra and boreal forest snow classes. Average end-of-winter SWE and winter precipitation are slightly higher at tundra sites (157 mm and 123 mm, respectively) than at boreal forest sites (145 mm and 114 mm, respectively). Snow sublimation flux equates to 16% of end-of-winter SWE and approximately 21% of the measured cumulative solid winter precipitation (Table 3).

Looking at water flux trends during the study period (2010–2021), the only significant relationship is an increase in winter solid precipitation at the lowland boreal sites (p value = 0.02 and $r^2$ = 0.39). There are no trends in the sublimation rates, SWE, or snow cover duration at the lowland boreal sites, nor any significant trends at the tundra sites.

**Table 3: Mean annual sublimation rates compared with snow cover duration, solid precipitation measurements, and snow water equivalent (SWE) measurements.**

| | Mean Sublimation ± SD[1] (mm yr$^{-1}$) | Snow Cover Duration ± SD (days) | SWE ± SD (mm) | Percent of SWE that Sublimates ± SD (%) | Solid Precipitation ± SD (mm) | Percent of Solid Precipitation that Sublimates ± SD[1] (%) |
|---|---|---|---|---|---|---|
| Tundra | 24 ± 10 | 254 ± 13 | 157 ± 29 | 16% ± 7% | 123 ± 31 | 20% ± 8% |
| Lowland Boreal | 21 ± 7 | 185 ± 20 | 145 ± 37 | 16% ± 7% | 114 ± 35 | 21% ± 10% |

**4.2.2 Water Vapor Fluxes**

Table 4 details mean annual sublimation in conjunction with other vapor fluxes (ET, condensation, and deposition) to show the relative magnitude of moisture transfer in two northern climatic and snow regions throughout the year.

While mean ET is the largest vertical vapor flux during the warm season (124 mm at tundra and 258 mm at boreal forest, Table 4), sublimation is still a substantial component of the winter water balance. Mean annual sublimation accounts for 8% of cumulative annual water vapor flux to the atmosphere (ET plus sublimation) at lowland boreal sites and 16% at tundra sites.

The relative importance of the downward fluxes varies, as condensation is minimal compared to ET (2% or less) while deposition is 15–20% of sublimation.

**Table 4: Sublimation rates and other mean annual water vapor fluxes measured by the eddy covariance sensors.**

| | Mean Sublimation ± SD (mm yr$^{-1}$) | Mean ET ± SD (mm yr$^{-1}$) | Mean Condensation ± SD (mm yr$^{-1}$) | Mean Deposition ± SD (mm yr$^{-1}$) |
|---|---|---|---|---|
| Tundra | 24 ± 10 | 124 ± 37 | 2.5 ± 1.2 | 6.0 ± 3.3 |
| Lowland Boreal | 21 ± 7 | 258 ± 39 | 5.1 ± 2.5 | 3.0 ± 1.0 |

**4.3 Differences in Annual Sublimation between Sites, Tree Presence, and Snow Classes**

Annual sublimation rates are grouped by site in boxplots in Figure 5. Welch's ANOVA and post hoc Games-Howell tests reveal that, on an annual scale, the lowland boreal bog (US-BZB) measures significantly lower sublimation rates than the

280 lowland boreal black spruce (US-BZS, p value = 0.03) and the tussock tundra (US-ICt, p value = 0.04). The remaining sites do not measure significantly different annual sublimation from each other.

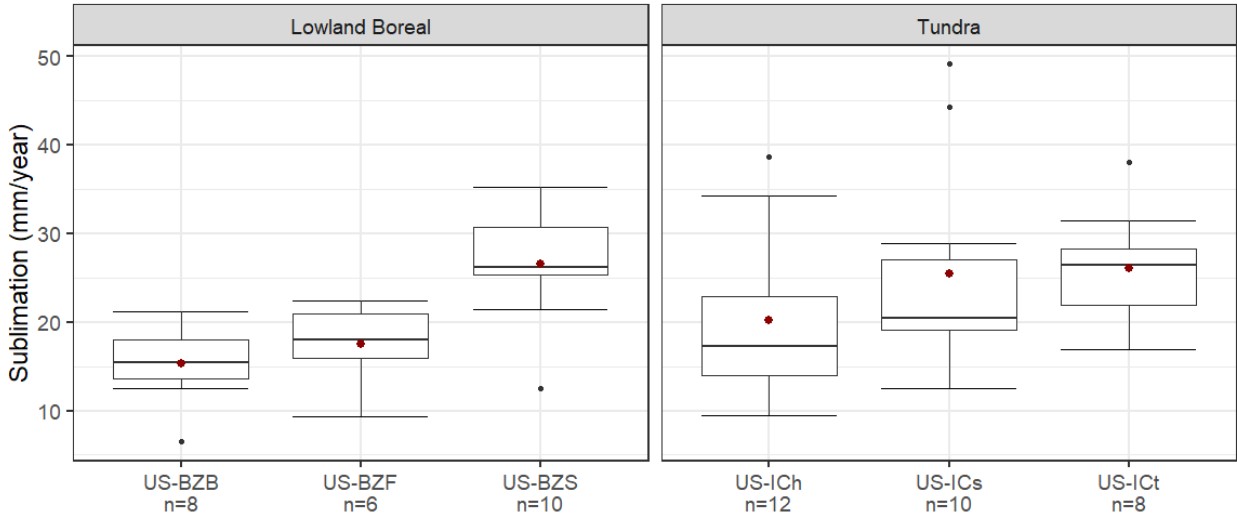

**Figure 5: Annual sublimation by site. Red dots represent the mean, boxes enclose the 1st and 3rd quartiles, horizontal line within the box is the median, whiskers denote the minimum value below the closest quartile ± 1.5 x interquartile range, and points outside the**
285 **whisker are considered outliers. US-BZB sublimation is significantly different from US-BZS and US-ICt.**

The Figure 6 boxplot pools sites with trees (US-BZS) and without trees (US-BZB, US-BZF, US-ICh, US-ICs, and US-ICt); this change in grouping aims to assess whether the canopy sublimation term differentiates a site's sublimation rates. The Welch's t-test demonstrates that sublimation rates are significantly different between sites with trees and without trees (p value = 0.02), a finding that is masked by the small site-to-site variation evaluated in Figure 5.

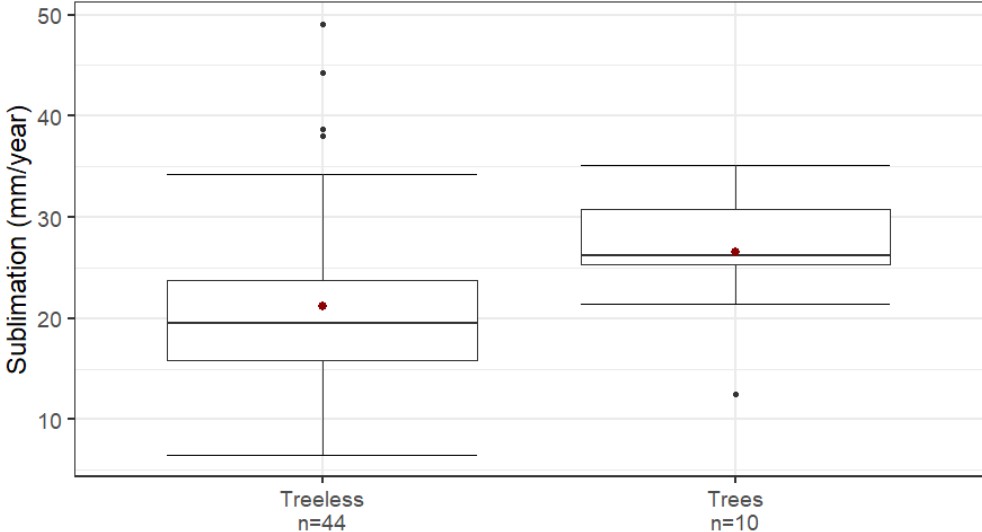

**Figure 6: Annual sublimation by tree presence. See Figure 5 caption for explanation of boxplot features. Annual sublimation at sites without trees are significantly different than the site with trees.**

Sites grouped by snow class measure insignificantly different annual sublimation rates between lowland boreal forest and tundra sites (p value = 0.24).

## 4.4 Meteorologic Drivers of Sublimation Rates

Table 5 contains the mean Pearson's correlation coefficients (r) and standard deviations between hourly and daily sublimation rates and meteorological variables at lowland boreal forest and tundra sites. The lowland boreal sites have stronger correlations with meteorological variables than the tundra sites. As seen in Table 5, the greatest disparities in the strength of relationship between regions are higher wind speed and lower relative humidity. Vapor pressure deficit (VPD) has the strongest relationship with daily sublimation at both lowland boreal (r = 0.69) and tundra (r = 0.40) sites. Other factors that promote greater sublimation rates include higher air temperature, elevated net radiation, and a low temperature gradient through the snowpack between the ground surface and station sensor.

Temperature gradient is inversely related to the air temperature measurements. Correlation and regression results show sublimation rates increase with higher air temperatures and lower temperature gradients (Table 5), and this disputes the authors' initial hypothesis that a larger temperature gradient through the snowpack could drive a water vapor flux towards the atmosphere.

All variables, except for net radiation and relative humidity, exhibit stronger correlations with daily summaries, likely due to reduced noise compared to the hourly data. However, the reduced statistical power for net radiation at the daily scale may result from the loss of meaningful information caused by aggregating daytime and nighttime values. It is unclear why relative humidity has weaker correlations at the daily scale at the tundra sites.

**Table 5: Hourly and daily sublimation mean Pearson's correlation coefficients (r) with standard deviations at the lowland boreal forest and tundra sites.**

|  |  | Hourly | Daily[1] |
|---|---|---|---|
| Air Temperature (°C) | Lowland Boreal | 0.40 ± 0.06 | 0.53 ± 0.05 |
|  | Tundra | 0.30 ± 0.04 | 0.40 ± 0.05 |
| Net Radiation (W m$^{-2}$) | Lowland Boreal | 0.43 ± 0.09 | 0.38 ± 0.16 |
|  | Tundra | 0.36 ± 0.14 | 0.37 ± 0.12 |
| Wind Speed (m sec$^{-1}$) | Lowland Boreal | 0.48 ± 0.06 | 0.56 ± 0.05 |
|  | Tundra | 0.20 ± 0.05 | 0.25 ± 0.03 |
| Vapor Pressure Deficit (hPa) | Lowland Boreal | 0.56 ± 0.10 | 0.69 ± 0.05 |
|  | Tundra | 0.33 ± 0.07 | 0.40 ± 0.08 |
| Temperature Gradient[2] (°C) | Lowland Boreal | -0.40 ± 0.05 | -0.47 ± 0.07 |
|  | Tundra | -0.28 ± 0.03 | -0.32 ± 0.03 |
| Relative Humidity (%) | Lowland Boreal | -0.44 ± 0.09 | -0.48 ± 0.07 |
|  | Tundra | -0.08 ± 0.02 | -0.05 ± 0.04 |

[1]Daily data are summarized as the mean value of all variables except for the sum of net radiation.

[2]Temperature gradient equals soil temperature minus air temperature.

Single and multiple linear regression ($r^2$) results between daily sublimation rates and meteorological variables are in Table 6. Single linear regressions with hourly and daily sublimation rates as the response variable show moderate relationships ($r^2 >$ 0.1) between air temperature, wind speed, net radiation, vapor pressure deficit, and temperature gradient (Table 6). Three patterns noted with the correlation coefficients hold:

1. wind speed and relative humidity at the lowland boreal sites have substantially stronger relationships with sublimation rates than at the tundra sites,

2. the lowland boreal sites have stronger trends with all meteorological variables than the tundra sites, and

3. the strength of the relationship of meteorological variables generally improves when the time scale is increased to daily summaries (except net radiation).

A forward stepwise analysis is performed to find the highest quality fully crossed MLR, which agreed at all sites, and includes air temperature, VPD, net radiation, temperature gradient, and wind speed. The MLR model explains 54–81% of the variance in daily sublimation and 43–62% of the variance in hourly sublimation rates, depending on the site.

A second MLR is included with more commonly measured meteorological variables – air temperature, wind speed, and relative humidity – and explains 26–69% and 17–53% of the variance in daily and hourly sublimations rates, respectively. The tundra sites have weaker relationships with this MLR than the lowland boreal sites, which is logical given the low-quality relationships with wind speed and relative humidity shown by single linear regression and correlation coefficient results (see pattern #1 listed above). An MLR with air temperature and wind speed explains 33–42% of the variance in the daily sublimation rates.

**Table 6: Single and fully crossed multiple linear regression mean coefficient of determination ($r^2$) results with standard deviation values between sublimation rates and meteorological variables at the lowland boreal forest and tundra sites. All p-values < 0.05.**

| | | Hourly | Daily[1] |
|---|---|---|---|
| Air Temperature (°C) | Lowland Boreal | $0.17 \pm 0.05$ | $0.28 \pm 0.05$ |
| | Tundra | $0.09 \pm 0.002$ | $0.16 \pm 0.04$ |
| Net Radiation (W m$^{-2}$) | Lowland Boreal | $0.19 \pm 0.08$ | $0.17 \pm 0.11$ |
| | Tundra | $0.17 \pm 0.08$ | $0.16 \pm 0.09$ |
| Wind Speed (m sec$^{-1}$) | Lowland Boreal | $0.23 \pm 0.06$ | $0.32 \pm 0.05$ |
| | Tundra | $0.03 \pm 0.01$ | $0.06 \pm 0.01$ |
| Vapor Pressure Deficit (hPa) | Lowland Boreal | $0.32 \pm 0.10$ | $0.48 \pm 0.07$ |
| | Tundra | $0.11 \pm 0.05$ | $0.17 \pm 0.07$ |
| Temperature Gradient[2] (°C) | Lowland Boreal | $0.16 \pm 0.04$ | $0.22 \pm 0.07$ |
| | Tundra | $0.08 \pm 0.02$ | $0.11 \pm 0.02$ |
| Relative Humidity (%) | Lowland Boreal | $0.20 \pm 0.07$ | $0.23 \pm 0.06$ |
| | Tundra | $0.00 \pm 0.00$ | $0.00 \pm 0.00$ |
| MLR: Air Temperature and Wind Speed | Lowland Boreal | $0.40 \pm 0.14$ | $0.53 \pm 0.04$ |
| | Tundra | $0.14 \pm 0.01$ | $0.23 \pm 0.02$ |
| MLR: VPD and Wind Speed | Lowland Boreal | $0.49 \pm 0.23$ | $0.64 \pm 0.16$ |
| | Tundra | $0.17 \pm 0.03$ | $0.24 \pm 0.07$ |
| MLR: Air Temperature, Wind Speed, Relative Humidity | Lowland Boreal | $0.53 \pm 0.18$ | $0.69 \pm 0.09$ |
| | Tundra | $0.17 \pm 0.01$ | $0.26 \pm 0.02$ |
| MLR: Air Temperature, VPD, Net Radiation, Temperature Gradient, Wind Speed | Lowland Boreal | $0.62 \pm 0.17$ | $0.81 \pm 0.05$ |
| | Tundra | $0.43 \pm 0.13$ | $0.54 \pm 0.09$ |

[1]Daily data are summarized as the mean value of all variables except for the sum of net radiation.

[2]Temperature gradient equals soil temperature minus air temperature.

Annual sublimation rates are proportional to the length of the snow cover season at all lowland boreal sites (Figure 7; all p-values < 0.05 and $r^2$ is 0.38–0.85), but there were no significant relationships between the sublimation rates and amount of solid precipitation or SWE. Sublimation rates at the tundra sites did not show significant relationship with the length of the snow cover season, solid precipitation, and SWE.

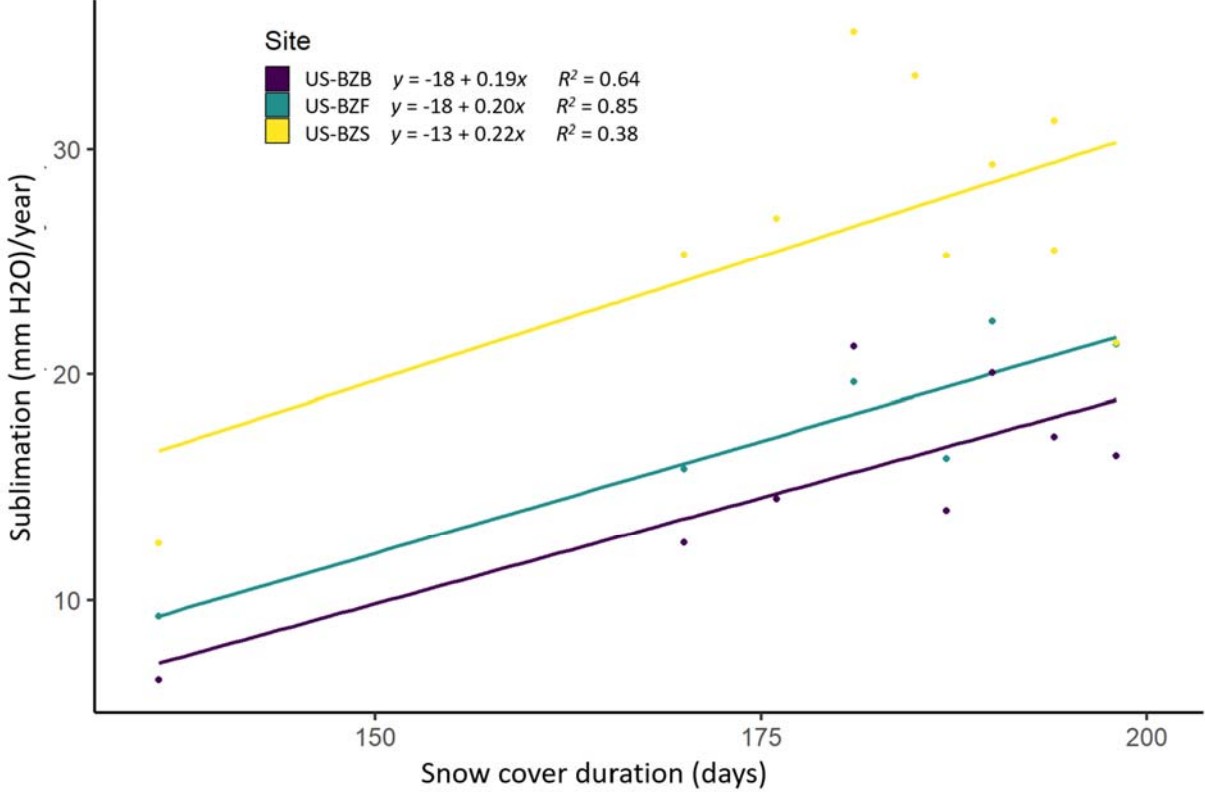

**Figure 7: OLS regression between annual sublimation rates and snow cover duration at the lowland boreal sites (p-values = 0.01–0.05 and r² = 0.38–0.85).**

**5 Discussion**

**5.1 Uncertainty of Sublimation Measurements**

EC measurements of sublimation contain three noteworthy types of uncertainty: (1) the fraction of total sublimation captured by EC method, (2) exclusion of evapotranspiration from sublimation estimates, and (3) inherent measurement errors.

First, the EC method primarily measures surface sublimation (Lackner et al., 2022; Marks et al., 2008; Reba et al., 2012; Sexstone et al., 2016; Stigter et al., 2018) and substantially lacks the blowing snow term due to 1) sublimation of snow crystals that occurs above the sensors in the suspension layer and 2) affected / lost measurements due to the obstruction of the optical path of the IRGA by suspended snow (Lackner et al., 2022). The literature is inconclusive as to the percentage of total sublimation measured by EC, which is related to site-specific conditions of the snowpack, wind regime, and upwind fetch (Reba et al., 2012). Recent results from a numerical model of drifting snow sublimation indicate that the dominant contributor of total sublimation occurs in the saltation layer (Wang et al., 2019), which is the near-surface layer where blowing snow particles move in a bounding motion over the snow surface (Kobayashi, 1972). These findings suggest that the EC method should be capable of quantifying most of the latent heat loss since the saltation layer is well below EC sensors at the study sites. Nevertheless, it is necessary for drawing appropriate conclusions that these findings be viewed as underestimates of total sublimation in environments prone to snow transport by wind (e.g., tundra snow).

Second, this study elected to calculate sublimation when snowpack is present, from snow cover onset in the fall to complete melt in the spring. During the melt season in the spring, it is possible that liquid water is present in the snowpack and, therefore, that some measurement periods were in fact evaporation of water, not sublimation of snow or ice crystals. This uncertainty when defining the measurement period results in a possible overestimate of sublimation rates during spring conditions.

Third, the uncertainty of latent heat (LE) measurements presents another source of error. A study in Maine, USA, quantified the uncertainty of EC flux data using two closely sited towers. LE uncertainty measured 5 Watts m$^{-2}$ averaged over the entire calendar year (Hollinger & Richardson, 2005; Hulstrand & Fassnacht, 2018). Study results also showed LE data are heterotactic; in other words, the uncertainty is not constant and increases with the magnitude of LE measurements.

**5.2 Evaluation of Sublimation Rates with Meteorologic and Environmental Data**

Correlation coefficients and linear regressions assess relationships between sublimation and meteorological variables to better understand drivers of sublimation. Hourly and daily sublimation rates exhibit moderate-quality significant relationships (Table 5 and Table 6): sublimation rates are positively correlated with air temperatures, VPD, net radiation, and wind speed, while negatively correlated with temperature gradient and relative humidity. The lowland boreal sites exhibit stronger relationships with meteorological variables than the tundra sites; the largest differences are associated with higher wind speeds and lower relative humidity (Table 5). The markedly higher winter wind speeds at the tundra sites (Table 1) may cause those EC sensors to capture a smaller proportion of total sublimation compared to the lowland boreal sites (see Section 5.1 discussion on underestimates during blowing snow events), and potentially affect the relationship between sublimation rates and wind speed and comparison across sites.

Stepwise selection of a MLR finds that sublimation rates have strongest relationship with a fully crossed model with air temperature, VPD, net radiation, temperature gradient, and wind speed, explaining 54–81% of the variance in daily sublimation. Most meteorological stations do not measure all these variables; therefore, a second MLR test is performed with common sensors (air temperature, wind speed, and relative humidity) to show how well these variables could explain sublimation rates with a simple model. The tundra sites' daily sublimation rates exhibit weaker relationships with these three variables (mean $r^2 = 26\%$) than the lowland boreal sites (mean $r^2 = 69\%$), which is congruent with the disparity between the two snow classes regarding the wind speed and relative humidity single linear regression and correlation coefficient results (Table 5 and Table 6). While the largest disparity is between wind speed and relative humidity, all meteorological variables show substantially stronger relationships with sublimation rates at the lowland boreal sites than at the tundra sites.

Annual sublimation has a positive significant relationship with snow cover duration at the lowland boreal sites (Figure 7). The snow cover duration relationship is logical: more days with snow present are more days that sublimation is possible. This relationship may be an important environmental driver of sublimation rates as climate changes. It is predicted that as Arctic and subarctic air temperatures continue to rise in the coming decades, the snow cover duration will decrease, though the amount of snow may increase (Bring et al., 2016; Brown et al., 2021; Thoman 2023), which was measured at the lowland boreal sites by a significant increase in solid precipitation during this 12-year study period.

Total sublimation is the sum of sublimation processes at the snowpack surface, in blowing snow, and in canopy-interception snow. This study demonstrates the importance of the canopy interception term in the total sublimation equation. Specifically, mean daily and mean annual sublimation rates measure highest at the site with a tree canopy (US-BZS; Table 2) and a Welch's ANOVA test indicates the US-BZS annual sublimation rates are significantly higher than the five other sites where all vegetation is low-stature and buried by snow throughout the winter (Figure 6). It is worth highlighting that US-BZS measures higher annual sublimation rates than the tundra sites, even though tundra sites have, on average, 69 additional days of snow cover per year (Table 2, Table 3, and Figure 5). These findings underscore the importance of the canopy term in the boreal forest snow class.

**5.3 Comparison of Sublimation Results with Other Studies**

The daily sublimation rates (0.08–0.15 mm day$^{-1}$) reported in this study are on the low end when compared with the reported rates found in the literature by the same method (see Table 7 for studies), though similar to those in northern climates. Nakai

et al. (2013) measured 0.09 mm day$^{-1}$ and 18.2 mm year$^{-1}$ in a 1-year study at a site within a lowland black spruce forest in Interior Alaska. Another study in the subarctic tundra of Hudson Bay (Lackner et al., 2022) measured 0.12 mm day$^{-1}$. These findings suggest that high latitude areas experience lower rates of daily sublimation than areas at lower latitudes in Table 7. Annual rates of sublimation are either not available or not included in the published research, but they would be an interesting comparison given the longer snow cover season at the Arctic and subarctic sites and additional days per year for sublimation to occur.

There are noteworthy differences between this study and some of those included in Table 7. Some studies took place for less than a snow season (Molotch et al., 2007; Pomeroy & Essery, 1999; Stigter et al., 2018). Sublimation rates from Pomeroy and Essery (1999) were measured during one season's blowing snow events only and are thus different from this study. Stigter et al. (2018) took measurements for 32 days in early winter months, rather than an entire snow season. Similarly, the study by Molotch et al. (2007) was confined to 40 days in the spring. Reba et al. (2012) only analysed sublimation for high-quality measurements and did not include gap-filled data, which left one site with as little as 16 days for the entire snow season from which to average. Furthermore, most of the studies (Knowles et al., 2012; Marks et al., 2008; Molotch et al., 2007; Reba et al., 2012; Sexstone et al., 2016; Stigter et al., 2018) took place at significantly lower latitudes (Table 7), where energy inputs are greater than in northern Alaska and the Arctic.

Vegetation is a factor that contributes to the comparatively low sublimation rates at treeless sites. The communities with vegetation completely buried under snow had significantly lower sublimation rates than the study site with trees (US-BZS; Figure 6). Vegetation that remains above the snowpack, such as trees and tall shrubs, increases total sublimation rates due to the addition of the canopy sublimation term (Molotch et al., 2007) or increases snow availability in sufficiently tall, exposed shrubs where blowing snow collects (Mahrt & Vickers, 2005).

**Table 7: Comparison of sublimation rates from all known studies that use the eddy covariance method.**

| Study | Site Type | Location | Latitude, Longitude | Sublimation (mm day$^{-1}$) |
|---|---|---|---|---|
| This study | Arctic tundra | North Slope, Alaska | 68°N,149°W | 0.08–0.10 |
| This study | Lowland boreal | Interior Alaska | 64°N, 148°W | 0.08–0.15 |
| Knowles et al., 2012 | Alpine tundra | Niwot Ridge, Colorado | 40°N, 105°W | 0.55 |
| Lackner et al., 2022 | Low-Arctic tundra | Hudson Bay, Canada | 56°N, 76°W | 0.12 |
| Marks et al., 2008 | Lodgepole pine forest | Fraser Forest, Colorado | 40°N, 106°W | 0.21 |
| Molotch et al., 2007 | Sub-alpine forest, below canopy | Niwot Ridge Forest, Colorado | 40°N, 105°W | 0.41 |
| Molotch et al., 2007 | Sub-alpine forest, above canopy | Niwot Ridge Forest, Colorado | 40°N, 105°W | 0.71 |
| Nakai et al., 1999 | Boreal forest, above canopy | Northern Japan | 42°N, 141°E | 0.6 |
| Nakai et al., 2013 | Lowland boreal | Interior Alaska | 65°N, 147°W | 0.09 |
| Pomeroy & Essery, 1999 | Prairie (blowing snow) | Western Canada | 52°N, 107°W | 1.8 |
| Reba et al., 2012 | Forest opening | Idaho | 43°N, 116°W | 0.39 |
| Reba et al., 2012 | Aspen forest | Idaho | 43°N, 116°W | 0.15 |
| Sexstone et al., 2016 | Forest opening | Colorado Rocky Mountains | 40°N, 105°W | 0.33–0.36 |
| Stigter et al., 2018 | Glacier | Himalaya | 28°N, 86°E | 0.99 |
| Stössel et al., 2010 | Alpine | Swiss Alps | 46°N, 9°E | 0.1 |

A few studies reported snow sublimation as the percentage of SWE lost to the atmosphere. Marks et al. (2008) measured 6.5% of total SWE sublimated over one snow season in Colorado. The study by Reba et al. (2012) occurred over two seasons and at two sites: a sheltered and an exposed site. Sublimation accounted for 4–8% and 16–41% of winter SWE, respectively. While Stigter et al. (2018) at the Yala Glacier in Nepal measured sublimation rates up to 30 times higher than in this study, the fraction of snowfall returned to the atmosphere was comparable, equalling 21% for the year investigated. Sublimation was only 5% of winter solid precipitation in a study conducted over three winters by Lackner et al. (2022) in the Canadian subarctic, though the authors noted that solid precipitation is high at the study site which could have lowered the percentage.

Some studies assessed the environmental controls on sublimation rates. Reba et al. (2012) determined that a strong vapor pressure difference, moderate wind speeds, high diurnal temperature range, and low relative humidity resulted in increased sublimation rates. Molotch et al. (2007) concluded that temperature gradients between vegetation and the snow surface and strong diurnal temperature fluctuations, both within the snowpack and near surface air, led to high vapor fluxes in the forested setting of his study site. Stigter et al. (2018) found that the best predictors of sublimation rates are VPD, wind speed, and air temperature using multiple regression analysis ($r^2 = 0.80$) for a glacial site in Nepal. The findings from these three studies and the present research (see Section 5.2) confirm that drivers of sublimation are complex and vary by region and site conditions, though recurring variables across studies include wind speed, air temperature, VPD.

Interestingly, evapotranspiration (ET) research at the same EC sites in Alaska found that ET at forest sites has a stronger relationship with relative humidity and wind speed than ET at tundra sites (Thungberg et al., 2021). These results are congruent with findings from the present study regarding sublimation, where both correlation coefficients (Table 5) and linear regressions (Table 6) show that the lowland boreal sites have a substantially higher dependence on relative humidity and wind speed relative to other variables compared with the tundra sites. The relationship between sublimation rates and wind speed is discussed in Section 5.2. However, no hypothesis is proposed for the weak relationship with relative humidity at the tundra sites.

**6 Conclusion**

Sublimation rates were computed for 12 water years in two distinct snow classes in northern Alaska – at three tundra sites and three lowland boreal forest sites. Mean surface sublimation rates range from 0.08–0.15 mm day$^{-1}$ and 15–27 mm year$^{-1}$, which is comparable to mean daily sublimation rates in northern regions reported by others using the same method (Lackner et al., 2022; Nakai et al., 2013).

There is substantial variability in annual sublimation rates between water years at all sites, with the standard deviation equal to nearly 50% of the mean. The lowland boreal sites have higher mean daily sublimation rates than the tundra sites, though the longer snow cover period on the tundra amounts to greater mean annual sublimation rates. Water vapor loss to the atmosphere is relatively steady throughout the winter months and peaks during spring months.

Annual sublimation rates significantly increase during years with longer snow cover duration at the boreal forest sites (no significant relationship at the tundra sites), a noteworthy finding as snow cover duration is shortening (Thoman 2023).

On average, approximately 21% of solid precipitation and 16% of SWE sublimate each year. Mean annual sublimation accounts for 8% of the cumulative mean annual water vapor flux to the atmosphere (ET plus sublimation) at lowland boreal sites and 16% in tundra. Our measurements confirm that sublimation is a substantial component of the annual water balance and that sublimation measurements contribute to an improved understanding of the regional winter hydrologic cycle.

The six northern Alaska EC towers used in this study present a unique dataset; the associated measurements are capable of quantifying surface sublimation at improved certainty than estimates made to date at these research sites. The measured

sublimation fluxes are representative of the global boreal forest and tundra seasonal snow classes as defined by Sturm and Liston (2021). These data can be used for diagnosing problems, improving, and validating turbulent fluxes in energy and mass balance models (Marks et al., 2008), which is particularly important under changing climate conditions that include modifications to high-latitude snow covers.

**Data and code availability**

Eddy covariance data from 2010 to 2021 are available through the AmeriFlux website (https://ameriflux.lbl.gov/sites/site-search/?availability) using AmeriFlux site ID listed in study area section.

Imnavait Creek SNOTEL daily precipitation data were downloaded from NRCS website (https://wcc.sc.egov.usda.gov/nwcc/site?sitenum=968) from 1981 to 2022.

The Imnavait Creek SWE data are available at the Arctic Data Center (https://arcticdata.io/catalog/view/doi%3A10.18739%2FA29G5GD77) from 1985 to 2017.

The UAF Imnavait Creek weather station data are available on the Water and Environmental Research Center website (https://ine.uaf.edu/werc/werc-projects/imnavait/current-data/meteorological-stations/imnavait-met/).

The SWE automated snow pillow data are available at the Bonanza Creek LTER website (https://www.lter.uaf.edu/data/data-detail/id/177).

Scripts used for analysis and plots are written in RStudio version 4.2.2. The code and datasets are available on GitHub (https://github.com/kstockert4/sublimation.git).

**Author contribution**

Kelsey Stockert performed the formal analyses and prepared the manuscript. Svetlana Stuefer and Eugénie Euskirchen supervised the project and reviewed and edited the manuscript. Svetlana Stuefer and Kelsey Stockert conceptualized this project and acquired the funding. Eugénie Euskirchen curated the eddy covariance data and provided technical expertise with that dataset.

**Acknowledgements**

Financial support for this project was provided by the National Institute of Water Resources (NIWR), part of the U.S. Geological Survey (USGS) Research Program 104(b) 2021, and National Aeronautics and Space Administration (NASA) awards 80NSSC21K1913 and 80NSSC21M0321. The data used in this project were obtained as part of a study funded by the National Science Foundation (NSF) Arctic Observing Network (AON) Project, "Collaborative Research: Tracking Carbon, Water, and Energy Balance of the Arctic Landscape at Flagship Observatories in Alaska and Siberia," Award 1936752, and publicly available by AON, the NSF's Arctic Data Center, AmeriFlux, and the Geophysical Institute at the UAF. This project was funded by NSF grants DEB LTREB 2011257 and DEB-0830997, as well as the USGS Climate R&D program. Special thanks to Lea Hartl from the Arctic Climate Research Center for assistance with solid precipitation data at Fairbanks. We thank the editor and Dr. Glen Liston, Dr. Steven Fassnacht, and one anonymous reviewer for insightful comments on earlier versions of this manuscript.

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
