# Peer review of "Sublimation Measurements of Tundra and Taiga Snowpack in Alaska"

_The Cryosphere, 2023_

## Author Comment (AC1)

**Response to RC1 comment on "Sublimation Measurements of Tundra and Taiga Snowpack in Alaska" by Kelsey Spehlmann, Eugénie Euskirchen, and Svetlana Stuefer.**

RC1: 'Comment on tc-2023-153', Steven Fassnacht, 27 Dec 2023

Dear Dr. Fassnacht,

We thank you for your constructive and insightful comments. The following pages contain comments that appear exactly as they were received. Our responses are inserted next to each comment in blue text. Thank you again for taking time to review our manuscript.

Sincerely,
Kelsey Spehlmann, Eugenie Euskirchen, and Svetlana Stuefer

General

As the authors illustrate through the literature that they cite, estimating sublimation is very important for the annual water balance, and few studies have examined multiple (>3) years, which they do.
Overall this is a good paper, but there are a number of steps that are not well explained and thus the methods are unclear.

Response 1: Thank you for pointing out that estimating sublimation is an important subject and that few papers estimate long-term sublimation fluxes. We appreciate suggestions on how to clarify methods and details of the analysis.

All the details are listed below; here are several examples, 1) 30% of the EC data are gap-filled. How? This is apparently in one, or all, of the Euskirchen papers;

Response 2: Additional information on data gaps and gap filling methodology were added to Section 3.1. "Sublimation calculations use both filtered latent heat measurements (70%) and gap-filled data (30%). Filtering primarily refers to removing data when there is optical impedance by precipitation or aerial contaminants. This is denoted by the automatic gain control values measured by the infrared gas analyzers. These values are used as a quality assurance/quality control variable for both flux and radiation data, with 60% as the maximum threshold AGC value. Data gaps occur from instrument malfunction, instrument calibration, or occasional power outages in winter months. For data gaps of 1–6 days, missing observations were replaced by the mean for that time period (half hour) and based on adjacent days using the ReddyProc software (Euskirchen et al. 2024). For data gaps of 1–2 weeks, marginal distribution sampling is used to fill missing data (Euskirchen et al., 2024)."

2) section 3.3 presents "standard" statistical methods to evaluate the relationship between sublimation rates and meteorological and environmental variables. To what end is this done? Is this the same as what is stated at the beginning of section 4.2?

Response 3: Section 3.3 has been re-written to better explain what was specifically done and why. "Standard statistical methods were applied to the dataset for the following analyses (Gottelli & Ellison, 2004): 1) Ordinary least squares (OLS) regressions were used to evaluate sublimation rates with other water fluxes, with meteorological and environmental variables, and over time; 2) Analysis of variance (ANOVA) calculated whether there are differences in sublimation rates between the six sites, between regions, and between sites with a canopy. For tests with more than two groups, the ANOVA is followed by post hoc Tukey test s for pairwise comparisons to identify which means among a set of groups are significantly different from each other; 3) and Pearson's correlation coefficient (r) and single (OLS) and multiple linear regressions (MLR) evaluated the relationship between sublimation rates and meteorological and environmental variables: Pearson's correlation coefficient (r) and single and multiple linear regression (MLR). The three methods use a significance level of 0.05."

We often do not have enough data, i.e., no EC measurements, to adequately estimate sublimation rates. The authors correlate EC-estimated sublimation with hourly and daily meteorological data (Air Temperature * VPD * Net Radiation * Temperature Gradient * Wind Speed). In Table 6, it looks as if these are multiplied together. The daily correlation is quite high (mean $R^2$ of 0.81 for the Lowland Boreal). The authors could consider doing a split sample analysis, i.e., leaving out 2 years to create the model, and then evaluating the model on the years left out. Further, are all five variables used in the MLR necessary? There can at least be a discussion of this.

Response 4: We used the * sign to indicate it is a fully crossed model, per methods in Gottelli and Ellison (2004). To clear confusion, we replaced the * with a comma and updated the Figure's caption to clarify fully crossed. We better describe model selection in Section 3.3 Statistical Methods and comment further in discussion. See also Response #6 regarding the MLR discussion.

Section 5.1 presents a discussion of uncertainty. The focus is underestimation due to blowing snow sublimation and "data processing." The former is informative. The latter is attributed to an overestimation of sublimation that is actually melt-evaporation. This is interesting. However, under the umbrella of "data processing," the authors should at least mention measurement errors and uncertainty (e.g., Hultstrand and Fassnacht, 2018; https://doi.org/10.1007/s11707-018-0721-0).

Response 5: Thanks for sharing this paper, we added this reference to Section 5.1.

The MLR (section 4.4) and section 5.2 present and discuss sublimation rates as a function of meteorological data. Some readers will not see the point of this analysis. I think that it is useful, as we often do not have EC data. However, we often have meteorological data from a regular weather station. The authors should consider using the bulk flux method as a comparison (as was done in various papers cited). At least provide a more thorough discussion of why the MLR is relevant here (and elsewhere). Also, consider which variables in the MLR are not readily available at a regular weather station, i.e., Temperature Gradient and perhaps Net Radiation. How would the MLR model degrade if only available at a regular weather station were used? This should computed (or at minimum discussed), since we typically don't have "Temperature Gradient," as defined in this paper.

Response 6: This a very good point, however the bulk flux model is beyond the scope of this study. This study took advantage of long-term EC measurements that had not been previously analyzed for sublimation. A future study of model-based methods with direct EC measurements could be of value. Further discussion has been added regarding stepwise explanation and the decrease in predictive power.

Specific
Line 22: "phase change from ice crystals in the snowpack" – technically they are snow grains and not ice crystals (see Fierz et al., 2009; IACS Guide, etc.)

Response 7: This sentence was updated to change "ice crystals" to "snow grains" and include the reference.

Line 30: "errors associated with solid precipitation measurements in the Arctic (Goodison et al., 1998)" – while this is a good reference. The authors should also consider the numerous recent papers (last decade) related to WMO-SPICE

Response 8: We added reference to Nitu et al. 2018.

In the Introduction, consider the paper by Herrero and Polo, 2016 The Cryosphere) as they also compare various sublimation estimation methods.

Response 9: Thank you for the reference. We added information from this reference into the fifth paragraph of the Introduction, with a list of other direct methods to estimate snow sublimation.

Line 33: "eddy covariance (EC) method … continuously measure latent heat fluxes" – this is mostly true. Sexstone et al. (2016) illustrated some of the uncertainty with EC estimates of the latent heat fluxes.

Response 10: For this and other reasons, "continuously measure latent heat fluxes" has been removed.

In Figure 1, if any of the photos correspond to the Ameriflux sites (Figure 2) add those labels at least to the captions, but preferably in the figures with "b)," etc.

Response 11: Ameriflux site IDs were added to both Figure 1 and caption.

Figures 1 and 2: perhaps combine these two figures as they both relate to the study sites

Response 12: This is a good suggestion that we also deliberated prior to manuscript submission, but decided that the photos and site locations are better shown as separate figures.

Lines 81-82: the "mean annual precipitation (MAP)" was estimated as "140–270 mm, with 60% of that occurring as snow" by Euskirchen et al. (2017). Without reading that paper, I am curious if/how the precipitation was adjusted for undercatch. This is relevant and should perhaps

be stated here. The estimate of precipitation impacts the computation of the % of sublimation to precipitation

Response 13: Added "for the years 1988-2007" to the lines to clarify that these values represent long term climate conditions. This is included in the Background section with the intention to add context to these regions. These values were not adjusted for undercatch, nor were they used in this manuscript's methods during computation of the % sublimation to precipitation. In fact, this is why we used the end-of-winter SWE, to serve as a proxy for more accurate cumulative solid precipitation.

Line 82: "air temperatures are below freezing" – use the term "colder than freezing" instead of "below freezing," as below has an elevational context

Response 14: Good catch. "colder than freezing" has replaced "below freezing" in this line.

Line 85, Line 122, and Figure 1f: How big are the "low stature" plants? I don't expect them to have any noticeable amount of canopy interception, but due to the thin snow cover (maximum of 50 cm), it is likely that the vegetation is exposed for at least part of the winter. How does this impact the aerodynamic characteristics across the snow surface?

Response 15: Added "(<0.5 m)" to this line. Furthermore, we believe the impact is minimal. There are not large woody shrubs. It is primarily grasses, sedges, small deciduous shrubs, and some forbs.

Line 88: instead of "can top 20 m s-1," use "can exceed 20 m s-1"

Response 16: "can top" replaced with "can exceed".

Lines 92-94: consider adding citations to these three sentences

Response 17: Two of the sentences were removed in the revised version of the manuscript.

Section 2.1: This is not crucial, but a monthly summary of mean temperature, total precipitation, mean and maximum wind speeds for the two sites (tundra and boreal) would be informative so understand the climate of the study sites

Response 18: Thank you for the suggestion. We added a table with monthly summaries of wind speed, air temperature, and precipitation to Section 2.1 to provide a comparison of meteorological settings between the tundra and boreal forest sites. This new table is shown below.

Table 1: Monthly meteorological summaries for tundra and lowland boreal regions.

| | Mean Daily Air Temperature (°C) | | Mean Daily Wind Speed (m s$^{-1}$) | | Max Daily Wind Speed (m s$^{-1}$) | | Total Precipitation Normal[1] (mm) | |
|---|---|---|---|---|---|---|---|---|
| | Tundra | Lowland Boreal | Tundra | Lowland Boreal | Tundra | Lowland Boreal | Tundra | Lowland Boreal |

| | | | | | | | |
|---|---|---|---|---|---|---|---|
| January | -18 | -20 | 2.6 | 0.9 | 22.7 | 12.6 | 9 | 15 |
| February | -17 | -16 | 2.9 | 1.1 | 20.1 | 8.3 | 13 | 13 |
| March | -16 | -10 | 2.6 | 1.3 | 15.9 | 8.8 | 9 | 10 |
| April | -9 | 1 | 2.5 | 1.5 | 12.7 | 8.5 | 10 | 9 |
| May | 0 | 11 | 2.4 | 1.5 | 14.5 | 7.6 | 18 | 14 |
| June | 8 | 16 | 2.5 | 1.4 | 12.2 | 6.2 | 46 | 38 |
| July | 10 | 17 | 2.4 | 1.3 | 26.9 | 21.4 | 80 | 57 |
| August | 6 | 13 | 2.3 | 1.2 | 11.2 | 6.3 | 72 | 53 |
| September | 0 | 7 | 2.3 | 1.1 | 14.4 | 8.8 | 33 | 34 |
| October | -7 | -1 | 2.3 | 1.0 | 12.9 | 13.0 | 23 | 19 |
| November | -15 | -12 | 2.6 | 1.0 | 19.5 | 18.4 | 14 | 19 |
| December | -18 | -16 | 2.4 | 0.9 | 15.9 | 10.8 | 12 | 14 |
| Annual | -6.3 | -0.8 | 2.5 | 1.2 | 16.6 | 10.9 | 339 | 295 |

Line 130 and prior: "gap-filled data (30%)" – this is a lot. The methodology used to fill in the gaps is apparently presented in one of the four papers by Euskirchen et al. (2012, 2014, 2017, 2020). However, since 30% are gap-filled, the method used needs to be presented in the paper, at least in an Appendix or Supplementary Information.

Response 19: Additional information on data gaps and gap filling methodology were added to Section 3.1. See Response #2.

Line 145: are station pressure data not required?

Response 20: Correct, station pressure data are not utilized in this study.

Lines 161-162: why are "[s]tandard statistical methods … applied to evaluate the relationship between sublimation rates and meteorological and environmental variables?" To what end. Explain what was specifically done. I assume that this is to compute the % of total or winter precipitation lost to sublimation? If not, then why is this done?

Response 21: Section 3.3 has been re-written to better explain what was specifically done and why. See Response #3.

Line 165: No one cares that Arc was used to create the maps, since spatial data are not used in the analysis

Response 22: Sentence removed.

Line 170-171: what is meant by "mean rates are 5–7% of the maximum daily rate?"

Response 23: Clarified by adding that the "mean *daily* rates are 5-7% of the maximum daily rate.

Figure 4: needs the same legend that is in Figure 3, unless you combine those figures (not necessary).

Response 24: Legend added to Figure 4.

Consider adding 2010 to Figure 4b, even with no results, so that the reader can visually line up the year across the two study domains.

Response 25: We modified Figure 4b to include 2010.

Consider putting the gridlines between the years, instead of the mid-point, so we know which bar belong to which year when there are sites missing for a specific year (e.g., Figure 4a 2014, 2015, 2016 and Figure 4b 2014 and 2016.

Response 26: The Figure 4 gridlines were modified.

As stated above, it would be useful to show some annual or winter summary data, at least peak snow depth, peak SWE and winter precipitation totals

Response 27: We added additional summary data to Section 2.1. See Response#18.

Lines 194-195: this sentence is part of the Methods and should be moved there

Response 28: The sentence was moved to Methods Section 3.3.

Lines 204-205 and the next sentence: how were the "winter solid precipitation increase[s] at the lowland boreal sites (p value = 0.02 and r2 = 0.39)" computed? Explain the method used

Response 29: We clarified this sentence by adding "as calculated by OLS regression". OLS regression has also been described as part of the Section 3.3 Statistical Methods (Response #3).

Table 2: the standard deviation (SD) seems large compared to the mean sublimation and % of solid precipitation sublimated. Are these distributions skewed, i.e., are the SD values misleading?

Response 30: We ran Shapiro Wilks tests on the mean sublimation and solid precipitation data at the tundra and lowland boreal sites. Results all came back to accept the null hypothesis that the data is normally distributed, or suggests there is simply not enough evidence to reject the null. We put a footnote in the table with Shapiro-Wilks test results and also provide the range of the data in addition to the standard deviation.

Table 3: consider adding a sentence that the condensation is minimal compared to ET, but that deposition (downward sublimation flux) is not minimal compared to sublimation (away).

Response 31: Added "The relative importance of the downward fluxes varies, as condensation is minimal compared to ET (2% or less) while deposition is 15–20% of sublimation." as the last sentence in Section 4.2.2.

Figure 5: consider adding what the other components of the box and whisker chart are, beyond the mean (red dot)

Response 32: Added the other components to the Figure 5 caption: "Red dots represents the mean, boxes enclose the 1st and 3rd quartiles, horizontal line within the box is the median, whiskers denote the minimum value below the closest quartile ± 1.5 x interquartile range, and points outside the whisker are outliers."

Table 4: From the text, these are correlations to daily sublimation. Add this to the caption.

Response 33: Added to the caption.

Table 4 "temperature gradient": needs to be explained better, as this is the gradient from the sensor through the air and the snowpack to the soil interface.

Response 34: Temperature gradient explanation added to the Methods in Section 3.3 and elaborated further in the results and discussion.

Citation: https://doi.org/10.5194/tc-2023-153-RC1

Response 35: Thank you again for taking time and providing comments on this sublimation manuscript.

---

## Author Comment (AC2)

**Response to RC2 comment on "Sublimation Measurements of Tundra and Taiga Snowpack in Alaska" by Kelsey Spehlmann, Eugénie Euskirchen, and Svetlana Stuefer.**

RC2: 'Comment on tc-2023-153', Anonymous Referee #2, 21 Jan 2024

Dear Reviewer,

We thank you for your constructive and insightful comments. The following pages contain comments that appear exactly as they were received. Our responses are inserted next to each comment in blue text. Thank you again for taking time to review our manuscript.

Sincerely,
Kelsey Spehlmann, Eugenie Euskirchen, and Svetlana Stuefer

General Comments:

The manuscript presents an interesting study on snow sublimation quantification in tundra and boreal forest sites in Northern Alaska. Snow sublimation was calculated using latent heat data from eddy covariance measurements over ~12 years at six sites differing by snow classes, vegetation communities, and permafrost. As stated in the manuscript, few studies have quantified sublimation in such environments by direct measurements. However, the knowledge gaps this study aims to address compared to previous studies, which are only briefly mentioned, need to be better highlighted in the Introduction.

Response 1: Thank you for your review and suggestions. We expanded our introduction by including references to the previous studies: "The winter water balance (in the absence of wind transport) is simple: snow water equivalent (SWE) equals precipitation minus sublimation (Liston & Sturm, 2004; Stuefer et al., 2020). But, in the Arctic, using field observations to make this moisture budget calculation produces sublimation estimates that are wide-ranging and unreliable due to errors associated with solid precipitation measurements (Goodison et al., 1998; Nitu et al., 2018). Systematic biases in solid precipitation measurements include wind undercatch, wetting loss, and evaporation loss (Fassnacht 2004; Goodison et al. 1998; Nitu et al. 2018). Sublimation can also be estimated by solving energy balance equations; including the Penman Monteith, bulk aerodynamic, and aerodynamic profile methods (Marks et al., 2008; Sexstone et al., 2016; Stigter et al., 2018) or direct measurements.

Next to quantifying the magnitude of snow sublimation, the study aimed to assess the sublimation spatial and temporal variability. This assessment can benefit significantly from the availability of the time series length at different sites. However, this variability is not enough described and discussed, for example among sites within the same snow class as shown by the mean monthly sublimation analysis. The relation between sublimation rates and snow cover length shows that the strength of the connection between the two variables can be quite different among sites for the same snow class, too. Differences in the interannual variability of cumulative annual sublimation between snow classes are also not described and further explored. Additional investigations and links to the effect of meteorological variables and, or site characteristics will be beneficial to improve this assessment.

Response 2: Thank you for this feedback. We have extended the discussion section to include these points. Specifically, we further assess the two snow classes presented (between and within) with regard to

(1) mean monthly sublimation rate variability, (2) cumulative annual sublimation rate interannual variability, (3) snow cover duration, (4) meteorological variables, and (5) site characteristics.

Similar considerations apply to the analysis between sublimation rates and meteorological drivers. No figures with the spatial and temporal variability of the meteorological data are shown, nor figures with correlations between variables. These figures need to be included to support the study results and discussion.

Response 3: The focus of this study is on sublimation rates; we therefore limited the bulk of the analysis to those rates. As a result, both the spatial and temporal variability as well as the correlations between variables of the meteorological data are not analyzed. To address this comment, we added monthly summaries of wind speed, air temperature, and precipitation to provide comparison of meteorological settings between the Tundra and Boreal Forest sites.

Table 1: Monthly meteorological summaries for tundra and lowland boreal regions.

| | Mean Daily Air Temperature (°C) | | Mean Daily Wind Speed (m s$^{-1}$) | | Max Daily Wind Speed (m s$^{-1}$) | | Total Precipitation Normal[1] (mm) | |
|---|---|---|---|---|---|---|---|---|
| | Tundra | Lowland Boreal | Tundra | Lowland Boreal | Tundra | Lowland Boreal | Tundra | Lowland Boreal |
| January | -18 | -20 | 2.6 | 0.9 | 22.7 | 12.6 | 9 | 15 |
| February | -17 | -16 | 2.9 | 1.1 | 20.1 | 8.3 | 13 | 13 |
| March | -16 | -10 | 2.6 | 1.3 | 15.9 | 8.8 | 9 | 10 |
| April | -9 | 1 | 2.5 | 1.5 | 12.7 | 8.5 | 10 | 9 |
| May | 0 | 11 | 2.4 | 1.5 | 14.5 | 7.6 | 18 | 14 |
| June | 8 | 16 | 2.5 | 1.4 | 12.2 | 6.2 | 46 | 38 |
| July | 10 | 17 | 2.4 | 1.3 | 26.9 | 21.4 | 80 | 57 |
| August | 6 | 13 | 2.3 | 1.2 | 11.2 | 6.3 | 72 | 53 |
| September | 0 | 7 | 2.3 | 1.1 | 14.4 | 8.8 | 33 | 34 |
| October | -7 | -1 | 2.3 | 1.0 | 12.9 | 13.0 | 23 | 19 |
| November | -15 | -12 | 2.6 | 1.0 | 19.5 | 18.4 | 14 | 19 |
| December | -18 | -16 | 2.4 | 0.9 | 15.9 | 10.8 | 12 | 14 |
| Annual | -6.3 | -0.8 | 2.5 | 1.2 | 16.6 | 10.9 | 339 | 295 |

Some variables, e.g. vapour pressure deficit and temperature gradient, need to be introduced, and the information they provide regarding processes at the different sites needs to be explained.

Response 4: We introduced VPD and temperature gradient in Section 3.2 and expanded on process-related explanations in the results and discussion.

The reasons why some correlations are more robust than others depending on the site and why correlation strength increases with the time scale should be adequately discussed.

Response 5: We added the explanation of correlation strength with time to the discussion. "Cumulative annual sublimation has a positive significant relationship with snow cover duration at the lowland boreal sites. The snow cover duration relationship is logical: more days with snow present are more days that sublimation is possible."

The discussion of the study's results compared to other studies is valuable. However, more clarity is necessary on how this study stands compared to the other studies, for example, in the vegetation and meteorological controls paragraphs.

Response 6: We clarified comparison with our study to other studies in the discussion. "Nakai et al. (2013) measured 0.09 mm day-1 and 18.2 mm year-1 in a 1-year study at a site within a lowland black spruce forest in Interior Alaska. Another study in the subarctic tundra of Hudson Bay (Lackner et al., 2022) measured 0.12 mm day-1. These findings suggest that high latitude areas experience lower rates of daily sublimation than areas at lower latitudes in Table 7. Longer snow cover seasons in Arctic tundra and boreal regions may lead to annual sublimation rates more comparable to those of lower latitudes; however, annual rates of sublimation in high latitudes are either not available or not included in the published research."

Consider addressing all the points mentioned above and in the following detailed comments to enhance the presented work's clarity, readability, and strength.

Detailed Comments:

Line 12: What does it mean "with differing permafrost conditions"? Please clarify.

Response 7: Replaced that phrase with "discontinuous permafrost and differing ecosystems" to distinguish the boreal forest sites from the continuous permafrost at the tundra sites.

Line 17-18: What is the key finding of this analysis? Please add.

Response 8: Added that "sublimation is a substantial component of the winter hydrologic cycle" into the abstract.

Line 25: Only one reference?

Response 9: The Stigter et al. (2018) reference provided that particular estimate of sublimation importance. Other papers have varying estimates.

Line 26: Please elaborate more on the challenges of measuring snow sublimation.

Response 10: The challenges of measuring snow sublimation is elaborated in the Introduction: "EC measurements are the most direct means available to measure vertical turbulent fluxes (Marks et al., 2008; Molotch et al., 2007; Reba et al., 2009, 2012; Sexstone et al., 2016; Stigter et al., 2018). However, EC towers that operate year-round are rare in much of Alaska due to challenges associated with the complexity and expense of maintenance during the harsh winter."

Line 30: Please elaborate more on the error associated with precipitation measurements.

Response 11: Added that "Systematic biases in solid precipitation measurements include wind induced undercatch, wetting loss, and evaporation loss (Fassnacht 2004; Goodison et al., 1998, Nitu et al., 2018)."

Line 34-35: Please add references.

Response 12: Added Guo et al. 2018; Herrero & Polo, 2016

Line 54: This could be moved in the study area section.

Response 13: We believe that the Introduction section benefits from noting the EC locations and associated snow classes in the study.

Line 69: Please mention what is Ameriflux.

Response 14: Clarification that "AmeriFlux is a network of EC research sites across the Americas" was added to Section 2.1.

Line 116: The function of this section here needs to be clarified. Types of sublimation can be mentioned in the Introduction, and details on the sublimation estimated by EC in section 3.1.

Response 15: This section is important part of study characteristics. It points out sublimation processes typical for our study locations: "Canopy sublimation takes place where snow is captured in tree canopies, but five of the six EC sites in this study are in low-growing vegetation environments where plants are completely covered by snow during the winter season so that the canopy sublimation term does not apply."

Line 127: If the instrument measures latent heat fluxes at 2.5-3m, is it below or above the vegetation canopy? In line 121, it is stated that the canopy sublimation term does not apply for five out of six EC sites; what about the site with tall vegetation? What is the effect of canopy sublimation there? Please clarify.

Response 16: This is a good point. The instruments are all above the canopy, as necessitated by the requirements of the eddy covariance methodology. We have added information specifying that the black spruce instrumentation is mounted higher – at 5 m. "EC towers at each of the six sites are equipped with a 3-D sonic anemometer and an infrared gas analyzer (IRGA) that measure the latent heat fluxes 10 times per second (10 Hz) 2.5–5 m above the ground (and above the canopy)." The effect of canopy sublimation is summarized in results, Section 4.3 Differences between Sites, Tree Presence, and Regions.

Line 129-130: This sentence needs to be clarified. What are filtered latent heat measurements and gap-filled data? How are gaps filled? It is not enough to mention that information is described in the references as this is the primary measuring method used. Please provide essential information.

Response 17: More detailed, essential information has been added to Section 3.1 regarding filtered and gap-filled data processing, such as "Filtering primarily refers to removing data when there is optical

impedance by precipitation or aerial contaminants. This is denoted by the automatic gain control values measured by the infrared gas analyzers. These values are used as a quality assurance/quality control variable for both flux and radiation data, with 60% as the maximum threshold AGC value."

Line 163: This variable has yet to be introduced; please specify what it is and which measure it provides.

Response 18: We added the vapor pressure deficit definition to Section 3.2.

Line 164-165: Please add references, explain the purpose of using a post hoc Tukey test and add the significance level.

Response 19: Included the Gottelli and Ellison (2004) reference, explained the Tukey test, and added the significance level for all statistical methods in Section 3.3: "Standard statistical methods were applied to the dataset for the following analyses (Gottelli & Ellison, 2004): 1) Ordinary least squares (OLS) regressions were used to evaluate sublimation rates with other water fluxes, with meteorological and environmental variables, and over time; 2) Analysis of variance (ANOVA) calculated whether there are differences in sublimation rates between the six sites, between regions, and between sites with a canopy. For tests with more than two groups, the ANOVA is followed by post hoc Tukey test s for pairwise comparisons to identify which means among a set of groups are significantly different from each other; 3) and Pearson's correlation coefficient (r) and single (OLS) and multiple linear regressions (MLR) evaluated the relationship between sublimation rates and meteorological and environmental variables: Pearson's correlation coefficient (r) and single and multiple linear regression (MLR). The three methods use a significance level of 0.05."

Line 177: Please clarify what is meant for "an average winter".

Response 20: Modified the sentence to read "over the snow season".

Line 180-181: What do these sublimation rates represent? The mean of the three sites? Please clarify.

Response 21: "*a range of* 1.5-2.4 mm/month…" added to the sentence to clarify.

Line 181-182: See previous comment.

Response 22: See Response 21.

Line 189-190: The relative change among a region's sites is quite variable over the years. Additional analyses are needed to support this statement.

Response 23: We removed those lines in the revised manuscript.

Line 220: Please specify whether significant differences refer to the mean of the annual sublimation rates at the different sites.

Response 24: We talk about cumulative annual rates (not mean of the annual sublimation rates). That line has been rephrased for clarity.

Line 231: Please state the two variables of the correlation. Please specify the type of correlation coefficient.

Response 25: Added the two variables (hourly and daily sublimation rates) as well as the type of correlation coefficient. Methods Section 3.3 also states that Pearson's correlation coefficient are used.

Line 251: Are the same explanatory variables providing the highest $r^2$ for the lowland boreal forest and tundra sites? It would be informative to show how the $r^2$ increases by including stepwise the explanatory variables.

Response 26: Yes, this is noted in Section 4.4 and further discussed in Section 5.2.

Line 235: This sentence needs to be clarified. Please refer to where information about "higher wind speed" and "lower relative humidity" can be found in the manuscript.

Response 27: We clarified this sentence and added reference to Table 4 in that sentence, where the reader can compare the r^2 values between sites for a given explanatory variable.

Line 272: Please elaborate more about the findings from the literature and this study.

Response 28: The intent of this line is to state how the scientific community does not have resolution on the matter and likely varies by site, as supported with numerous references (Lackner et al., 2022; Marks et al., 2008; Reba et al., 2012; Sexstone et al., 2016; Stigter et al., 2018; Wang et al. 2019).

Line 285-286: This sentence needs to be clarified. How is it possible to deduce this statement from Tables 4 and 5?

Response 29: Thank you for pointing this out. We restated the sentence to state that sublimation rates were positively correlated with air temperatures, VPD, and net radiation and negatively correlated with temperature gradient, wind speeds, and relative humidity.

Line 301: Which lower latitudes? Please clarify. This terminology is used in line 312, too, making the context unclear.

Response 30: Latitude is in Table 6: Comparison of sublimation rates from all known studies that use the eddy covariance method (please see fourth column).

Line 309-310: This sentence needs to be clarified.

Response 31: The sentence has been rephrased to clarify what high-quality measurements are.

Line 319-320: Is it not possible to advance any explanation about these results?

Response 32: This sentence has been removed.

Figure 4: Legend is missing; please add it.

Response 33: Thank you, the legend was added.

Figure 5: Please add some explanation in the caption about significant differences.

Response 34: The explanation was added: "Cumulative annual sublimation by site. Red dots represent the mean, boxes enclose the 1st and 3rd quartiles, horizontal line within the box is the median, whiskers denote the minimum value below the closest quartile $\pm$ 1.5 x interquartile range, and points outside the whisker are outliers. US-BZB sublimation is significantly different from US-BZS and US-ICt."

Figure 6: In the manuscript, the range reported for $r^2$ is 0.4-0.8. Please check consistency. Please add which $r^2$ belongs to which relation and define what $r^2$ is in the caption.

Response 35: We fixed this inconsistency and appreciate you noting the mistake.

Table 5: Please define $r^2$ in the caption.

Response 36: Added to the caption "coefficient of determination (r^2)"

**Citation**: https://doi.org/10.5194/tc-2023-153-RC2

Response 37: Thank you again for taking time and providing comments on this sublimation manuscript.

---

## Author Response (AR2)

**Response to RC2 comment on "Sublimation Measurements of Tundra and Taiga Snowpack in Alaska" by Kelsey Spehlmann, Eugénie Euskirchen, and Svetlana Stuefer.**

RC2: 'Comment on tc-2023-153', Anonymous Referee #2, submitted on 19 Jun 2024

*Dear Reviewer,*

*Thank you for your thoughtful review, comments, and suggestions. We are happy to answer your questions and provide more information.*

*The following pages contain comments that appear exactly as they were received. Our responses are inserted next to each comment in blue text. Thank you again for taking time to review our manuscript.*

*Sincerely,*
*Svetlana Stuefer on behalf of the coauthors*

**General comments:**
The manuscript shows significant improvements compared to the previous version. While the authors have addressed most of the comments, the following must be considered. The authors were invited to consider additional investigations to assess whether the snow sublimation variability among sites within the same snow class or among years could be related to specific meteorological conditions and site characteristics. This point still needs to be addressed. Table 1 is perfect for characterizing, on average, the meteorological conditions in the tundra and lowland boreal regions over the entire study period. Still, it does not support explanations for the snow sublimation variability mentioned above.

*Response 1: We appreciate Reviewer's suggestion to perform additional investigation to access if there is a statistically significant relationship between sublimation rates and specific meteorological conditions that can explain variability in sublimation rates. We address Reviewer's point by stating the following:*

1) *"Single linear regressions with daily sublimation rates as the response variable show moderate relationships ($r^2 > 0.1$) between air temperature, wind speed, net radiation, vapor pressure deficit, and temperature gradient." (Line 310–311, Table 5)*
2) *"Cumulative annual sublimation rates are proportional to the length of the snow cover season at all lowland boreal sites …, but there were no significant relationships between the sublimation rates and amount of solid precipitation or SWE. Sublimation rates at the tundra sites did not show significant relationship with the length of the snow cover season, solid precipitation, and SWE". (Line 331–334)*
3) *See Response 3 for details on MLR analysis and Table 6.*

*We would like to use this response as an opportunity to emphasize that the main goal of our paper is to calculate and quantify sublimation rates in the Arctic tundra and subarctic environments. The quantification of snow sublimation rate provides novel contribution that has not been done before. While statistical analysis presents an interesting exploration into the potential relationships with weather variables, the main value of this paper is in accurately quantifying the snow sublimation rates in remote and harsh environments.*

Although asked, figures showing the correlations between sublimation rates and meteorological variables for the lowland boreal and tundra are not displayed. How the dataset is used in the different analyses needs to be added in the section on methods since some analyses are performed with the cumulative annual snow sublimation rates and others with hourly and daily sublimation rates over the entire study periods, some considering the six sites and others with the data aggregated in lowland boreal and tundra regions.

*Response 2: We disagree with Reviewer's statement. Information on correlations between sublimation rates and meteorological variables for lowland and tundra is clearly presented in the manuscript in Table 5 "Hourly and daily sublimation mean correlation coefficients (r) with standard deviations at the lowland boreal forest and tundra sites". The data used (hourly or daily) are also clearly stated in the table.*

The response clarified that the highest coefficient of determination of the multiple linear regression resulted from a stepwise approach, although the different steps are not shown. Showing these steps would highlight how the explanatory power of the relation increases or degrades with the addition of the meteorological variables and demonstrate whether or not the increase is significant between the regions and among the sites of the same region. The description of how the multiple linear regression approach is applied needs to be improved in the section on the methods.

*Response 3: Additional details on regression models were added to the method section. We clarified how the meteorological variables were introduced in the stepwise regressions and the effect it had on the regression models. The revised text in methods section now reads (Line 209–213): "The meteorological variables were added in the stepwise regression in the following order: 1) air temperature and wind speed, 2) vapor pressure deficit (VPD) and wind speed, 3) air temperature, wind speed, and relative humidity, 4) air temperature, VPD, net radiation, temperature gradient, and wind speed. Regression models were evaluated based on their p-values, $r^2$, and adjusted $r^2$." A sentence on a MLR analysis with two meteorological variables was added in the result section (Line 324–325): "An MLR with air temperature and wind speed explains 33–42% of the variance in the daily sublimation rates."*

In addition to the sole SOS Project, 2023, other studies on the impact of snow sublimation on water balance should be cited. This reference needs to be double-checked, as the link in the Bibliography leads to a page not found.

*Response 4: The link to SOS project 2023 was removed. A recently published article by Lindquist and colleagues was added to the reference list: Lundquist, J. D., Vano, J., Gutmann, E., Hogan, D., Schwat, E., Haugeneder, M., Mateo, E., Oncley, S., Roden, C., Osenga, E., and Carver, L: Sublimation of Snow, Bull. Am. Meteorol. Soc., 105(6), doi: 10.1175/BAMS-D-23-0191.1, 2024.*

**Detailed Comments:**
Line numbers refer to the marked version.
Line 159: It is clear from the response, but not reading the section. Please rephrase to make it understandable that it is an integral part of the study characteristics.

*Response 5: Line 159 in marked manuscript is the sub-section title 2.2 Types of Sublimation. The print screen of the manuscript shows text in Line 159.*

**2.2 Types of Sublimation**

160    Total sublimation equals st

Lines 220-224: Please provide more information on the analysis of variance, the type of test, what it tests for, and why it was chosen compared to other statistical tests.

*Response 6: We specified that a one-way Welch's t-test was applied to investigate whether there are significant differences in sublimation rates across the six sites, between regions/snow classes, and between sites with and without a canopy (Line 193). The t-test is particularly useful for small sample sizes and remains a robust option for central tendency comparisons.*

Line 316-317: Please add "lowland boreal forest" and "tundra" so it is clear what you mean by "grouped by region/snow class".

*Response 7: The suggested wording was added. The revised sentence now reads (Line 289–290): "Sites grouped by snow class measure insignificantly different cumulative annual sublimation rates between lowland boreal forest and tundra sites (p value = 0.24)."*

Line 330: "Temperature gradient measures highest in the coldest months (November–March)…" please add a reference to a table or figure.

*Response 8: The statement in question is removed.*

Lines 362: Please clarify what does it mean "Sublimation rates at the tundra sites are not measurably affected by the same variables".

*Response 9: The sentence in question is revised to state (Line 334): "Sublimation rates at the tundra sites did not show significant relationship with the length of the snow cover season, solid precipitation and SWE."*

Lines 420-422: Please add the reference to a table or figure where "US-BZS shows significantly higher annual sublimation rates than the tundra sites".

*Response 10: We added "(Table 2, Table 3, and Figure 5)" to the end of the sentence to refer to the summary statistics of each site (Table 2), the snow cover duration in each region (Table 3), and the boxplot comparison (Figure 5). The revised sentence now reads (Line 386): "It is worth highlighting that US-BZS measures higher annual sublimation rates than the tundra sites, even though tundra sites have, on average, 69 additional days of snow cover per year (Table 2, Table 3, and Figure 5)."*

Figure 2: Please add "Imnavait Creek Station" and "Fairbanks Station" to show where the precipitation in Table1 is measured.

*Response 11: Revised Figure 2 includes reference to the Imnavait Creek stations. Fairbanks station is outside of the map on Figure 2b. The exact location of Fairbanks Station is added in the data section: "Solid precipitation is not measured at boreal forest sites; therefore, we used precipitation measurements from the NOAA, NWS (National Oceanic and Atmospheric Administration, National Weather Service) weather station USW0002641 at Fairbanks International Airport at 64°48'N, 147°52'W (https://www.ncdc.noaa.gov/cdo-web/datasets/GHCND/stations/GHCND:USW00026411/detail)." (Line 185)*

Figures 3 and 4: Please adopt consistent units of mm/year

*Response 12: Revised Figures 3 and 4 now have units of mm/month and mm/year.*

Figure 5, 6, 7: Please adopt consistent units of mm/year

*Response 13: Revised Figures 5–7 now have units of mm/month and mm/year.*

Figure 6: This figure shows that the two groups have different sample size. What is the effect on the test results? The results of the post-hoc Tukey tests in section 4.3 and Figure 5 show that the cumulative annual sublimation at sites with trees (US-BZS) is not significantly different from those at the tundra sites without trees. This finding suggests that the effect of the canopy sublimation term makes a difference within the lowland boreal region but not beyond. Can you really draw strong conclusions?

*Response 14: We agree two groups have different sample size. Methods were adjusted (Section 3.3 updated) to run a Welch's t-test to accommodate the unequal group sizes, which ensures valid comparisons between group means. The Welch's t-test confirms that sublimation rates at sites with a canopy are significantly different than sites without a canopy. To address the Reviewer's note on contradiction between findings in Figure 5 and Figure 6, we revised Line 281–284 to clarify the meaning of pooling data in different ways: "The Welch's t-test demonstrates that sublimation rates are significantly different between sites with trees and without trees (p value = 0.02), a finding that is masked by the small site-to-site variation evaluated in Figure 5."*

Table 1: Please specify in the caption if air temperature, mean, and max wind speed values are averages at the three sites within each region.

*Response 15: We added "Mean air temperature, mean wind speed, and max wind speed are averages of the three sites in each region." to the caption (Line 90).*

Table 2: Please add over which period the daily and annual sublimation rates are calculated.

*Response 16: We included "The daily and annual sublimation rates are calculated for water years with complete records." to the caption (Line 226).*

Table 4: Please specify Pearson's correlation coefficients in the caption.

*Response 17: We added "Pearson's" correlation coefficients to the caption (Line 268).*

---

## Author Response (AR3)

29 January 2025 Dear Editor and Reviewer,

Thank you for your thoughtful review, comments, and suggestions. We are happy to answer your questions and provide more information.

The following pages contain comments that appear exactly as they were received. Our responses are inserted next to each comment in blue text. Thank you again for taking time to review our manuscript.

**Sincerely,**

Svetlana Stuefer and Kesley Stockert on behalf of the coauthors

22 Dec 2024 Editor decision: Publish subject to minor revisions (review by editor) by Andrea Popp

Dear Kelsey Stockert and co-authors,

We finally received the comments from the two reviewers. While the reviewers acknowledge the improvements you made, referee#2 has highlighted remaining issues that need to be addressed before the manuscript can proceed.

Please clarify and expand particularly on the reviewer's points regarding 1) the variability in snow sublimation across sites and years, considering site characteristics and meteorological conditions,

Response 1: We appreciate the Editor's assessment of the manuscript and comments on improvements. We hope that the bolded text in this response makes clear how our revisions and clarifications address your request.

The variability in snow sublimation across sites and years is expanded in section 4.1 by clarifying and revising following sentences to address the **variability in sublimation across sites and years**: Lines 243–248 "Broadly, tundra site sublimation rates increase from US-ICh (ridge) to US-ICs (valley bottom) to US-ICt (mid-slope). At the lowland boreal sites, sublimation rates are greatest at US-BZS (black spruce) and lowest at US-BZB (bog). There is high interannual variability in sublimation rates (Figure 4). The most extreme range at US-ICs measures a nearly 40 mm year-1 difference of sublimated water between 2015 and 2021. The relative inter-site rates between sites variability in sublimation rates within tundra and lowland boreal forest snow classes is lower than interannual variability (Figure 4)."

Section 4.3 tests the statistical significance of the variability in snow sublimation across sites and **site** *characteristics* in

- Figure 5 shows that "... the lowland boreal bog (US-BZB) measures significantly lower cumulative annual sublimation rates than the lowland boreal black spruce (US-BZS, p value = 0.03) and the tussock tundra (US-ICt, p value = 0.04). The remaining sites do not measure significantly different cumulative annual sublimation from each other."
- 2) Figure 6 shows another boxplot that "pools sites with trees (US-BZS) and without trees (US-BZB, US-BZF, US-ICh, US-ICs, and US-ICt)...[and] demonstrates that sublimation rates are significantly different between sites with trees and without trees (p value = 0.02), a finding that is masked by the small site-to-site variation evaluated in Figure 5.

Section 4.4 contains results for correlations (Table 5) and single and multiple linear regressions (Table 6) to evaluate the effects **meteorological conditions** have on sublimation rates/variability.

2) the use of datasets in statistical analyses to ensure the methods clearly state which datasets were used and for what purposes,

Response 2: Section 3.3 was re-written to clarify the use of the data and methods (Lines 190–206). Most notably the addition of Lines 214–220: "data are summarized at different time scales: hourly, daily, monthly, and annual. Reporting daily rates is valuable for comparison with findings from other studies in the literature (Section 5.3). Correlations and regressions with meteorological variables are conducted using the hourly and daily data because these relationships are stronger at finer temporal resolutions. In contrast, regressions with environmental variables, namely snow cover duration, SWE, and solid precipitation, are meaningful (and available) only at the annual scale. ANOVA tests are applied to annual data to provide a clearer understanding of the impacts of sublimation over entire winters; and avoid the limitations of daily rates, which fail to capture the substantial difference in the snow class's snow cover duration (see specifics of differences in Section 4.2.1)."

and 3) a discussion of correlations at different temporal resolutions (hourly vs. daily) and their implications.

Response 3: A discussion was added to Section 4.4 (Lines 308–311): "All variables, except for net radiation and relative humidity, exhibit stronger correlations with daily summaries, likely due to reduced noise compared to the hourly data. However, the reduced statistical power for net radiation at the daily scale may result from the loss of meaningful information caused by aggregation daytime and nighttime values. It is unclear why relative humidity has weaker correlations at the daily scale at the tundra sites."

Furthermore, additions to Section 5.2 (Lines 373–378): "The lowland boreal sites exhibit stronger relationships with meteorological variables than the tundra sites; the largest differences are associated with higher wind speeds and lower relative humidity (Table 5). The markedly higher winter wind speeds at the tundra sites (Table 1) may cause those EC sensors to capture a smaller proportion of total sublimation compared to the lowland boreal sites (see Section 5.1 discussion on underestimates during blowing snow events), and potentially affect the relationship between sublimation rates and wind speed and comparison across sites."

Please revise the manuscript to address all comments of referee#2. If you choose not to implement specific suggestions, provide a clear justification in your response letter.

Thank you for addressing these points. I look forward to your updated submission.

Best regards, Andrea Popp

**Response to RC2 comment on "Sublimation Measurements of Tundra and Taiga Snowpack in Alaska" by Kelsey Spehlmann, Eugénie Euskirchen, and Svetlana Stuefer.**

RC2: 'Comment on tc-2023-153', Anonymous Referee #2, submitted on 20 Dec 2024

**General comments:**

The manuscript has significantly improved compared to the previous version and a few comments and clarification issues remain.

**Thank you for your review.**

In my previous review comments, the authors were invited to consider additional investigations to assess whether the snow sublimation variability among sites within the same snow class or among years could be related to specific meteorological conditions and site characteristics. In addition to vegetation the six sites are described to have different topographic positions and expositions. The authors did not fully address this point as 1), 2) for SWE, and 3) in the response relate to regressions over the entire study period and combined in lowland boreal forest and tundra. Additional investigations would have also more strongly supported the conclusions on the snow sublimation differences between sites, tree presence, and snow classes.

Response 1: We agree that variability in snow sublimation between sites, tree presence, and snow classes is an important aspect of this paper. An additional analysis for correlations between sublimation and specific meteorological variables at individual sites (tundra ridge, tundra tussock, tundra fen, boreal bog, boreal fen, boreal black spruce) was performed during earlier stages of manuscript development and preparation. At that time, we noticed the greater influence of snow-climate conditions over individual site meteorological conditions on sublimation rates in our study sites. This is because sites within the same snow class are located only 0.5 km apart with similar weather conditions, whereas the snow classes are over 600 km and represent distinct climate zones (Arctic Alaska vs Interior Alaska). We added clarification in Lines 221–223.

Similarly, summary statistics of the wind speed data from individual sites show that difference in wind

*speed between climate zones* is more pronounced than difference between individual sites, i.e. tundra climate is substantially winder than the lowland boreal climate. The wind speed plot provides additional analyses on the difference in topographic positions and expositions and supports our approach for aggregating data by snow class during data analysis and synthesis.

In addition, we clarified (Line 94) that EC towers used in this study were originally established to understand carbon and water cycling in distinct northern ecosystems (rich fen, bog, peat plateau, wet sedge, tussocks, dry heath). These distinct biological and soil criteria were the driving factors for selecting location of the individual sites.

With regards to **SWE differences between individual EC towers**, there are no corresponding SWE measurements at individual sites. We used SWE measurements available at the nearest location. Furthermore, SWE is only collected as a single end-of-winter measurement and cannot be reduced to evaluate hourly or daily data. Please, also see Response #1 to the Editor.

In my previous review comment on how the dataset was used in the different statistical analyses, the point was that the results of some statistical analyses, e.g. regressions and correlations with meteorological variables, are presented for hourly and daily data as classes (boreal and tundra), others, e.g. regressions with snow cover duration and analysis of variance, for annual data and distinct sites. The methods should clearly state which type of data sets was used for which statistical analysis and purpose.

Response 2: Section 3.3 was re-written to clarify the use of the data, most notably the addition of Lines 214–223: "... data are summarized at different time scales: hourly, daily, monthly, and annual. Reporting daily rates is valuable for comparison with findings from other studies in the literature (Section 5.3). Correlations and regressions with meteorological variables are conducted using the hourly and daily data because these relationships are stronger at finer temporal resolutions. In contrast, regressions with environmental variables, namely snow cover duration, SWE, and solid precipitation, are meaningful (and available) only at the annual scale. ANOVA tests are applied to annual data to provide a clearer understanding of the impacts of sublimation over entire winters; and avoids the limitations of daily rates, which fail to capture the substantial difference in the snow class's snow cover duration (see specifics of differences in Section 4.2.1). Lastly, most analyses group the data by snow class. This approach reflects the greater influence of snow-climate conditions over individual site meteorological conditions on sublimation rates. Sites within the same snow class are located only 0.5 km apart with similar weather conditions, whereas the snow classes are over 600 km and represent distinct climate zones (Shulski & Wendler, 2007)."

The results show that some correlations increase at daily resolution for specific classes and meteorological variables, while they remain low despite the temporal scale for others. These findings should be discussed.

Response 3: Additional discussion was added to Section 4.4 (Lines 308–311): "All variables, except for net radiation and relative humidity, exhibit stronger correlations with daily summaries, likely due to reduced noise compared to the hourly data. However, the reduced statistical power for net radiation at the daily scale may result from the loss of meaningful information caused by aggregation daytime and nighttime values. It is unclear why relative humidity has weaker correlations at the daily scale at the tundra sites."

We also added to the Discussion Section 5.2 (Lines 373–378): "The lowland boreal sites exhibit stronger relationships with meteorological variables than the tundra sites; the largest differences are associated with higher wind speeds and lower relative humidity (Table 5). The markedly higher winter wind speeds

at the tundra sites (Table 1) may cause those EC sensors to capture a smaller proportion of total sublimation compared to the lowland boreal sites (see Section 5.1 discussion on underestimates during blowing snow events), and potentially affect the relationship between sublimation rates and wind speed and comparison across sites."

Lastly, we elaborated on a point in Section 5.3 (Lines 443–447): "correlation coefficients (Table 5) and linear regressions (Table 6) show that the lowland boreal sites have a substantially higher dependence on relative humidity and wind speed relative to other variables compared with the tundra sites. The relationship between sublimation rates and wind speed is discussed in Section 5.2. However, no hypothesis is proposed for the weak relationship with relative humidity at the tundra sites."

The results of the linear regressions between sublimation rates and single meteorological variables with hourly data are reported in the table but not mentioned in the related section.

Response 4: We revised this sentence to include hourly data. The revised sentence now reads: "Single linear regressions with **hourly and** daily sublimation rates as the response variable show moderate relationships (r2 > 0.1) between air temperature, wind speed, net radiation, vapor pressure deficit, and temperature gradient (**Table 6**)."

Differences between hourly and daily results should be all reported and discussed.

Response 5: We revised Lines 323–324: "...the strength of the relationship of meteorological variables generally improves when the time scale is increased to daily summaries (except net radiation)." Please also see the added discussion noted in Response 3.

On the point of multiple linear regression, it is necessary also to show the coefficient of determination values in Table 6 for 1) air temperature and wind speed, 2) vapour pressure deficit (VPD) and wind speed as a result of the stepwise approach for the information completeness.

*Response 6: Table 6 was revised to include MLR results for 1) air temperature and winds and 2) VPD and wind speed.*

Detailed comments: Line 100: "occurring as snow" Please specify from and to which month for lowland boreal sites.

Response 7: See Lines 84-85 "Mean daily air temperatures remain below freezing from October to April in tundra sites and from October to March in lowland boreal forest sites (Table 1)" and Lines 276–278 "Snow cover duration at the tundra sites is approximately two months longer than at the lowland boreal sites. On average, snow cover duration is 254 days at tundra sites (mean date of snow onset is **September 19 and snow melt is June 1**) and 185 days at boreal sites (mean date of snow onset is **October 19 and snow melt is April 22**; Table 3)."

Line 190: Please specify which are the environmental variables and the other water flux.

Response 8: We clarified Section 3.3 (Lines 192–196): "The magnitude of daily, monthly, and annual sublimation rates are calculated for water years with complete records. Mean values, standard deviation, and standard error of the mean are used to compare the variability in sublimation rates between sites (Section 4.1), to compare sublimation with environmental conditions (snow cover duration, SWE, solid precipitation) (Section 4.2.1), and to evaluate sublimation rates with other water fluxes (ET, condensation, and deposition) (Section 4.2.2).

Line 191: Please specify what "over time" means, e.g., over the water year with complete records.

Response 9: The sentence in question was removed.

Line 221: Please define snow cover duration with the other environmental variables as it suddenly appears in the results and how snow cover duration, snow onset and melt are calculated.

Response 10: Definition of snow cover duration was added (Line 164): "... snowpack presence **and snow** cover duration are determined from the albedometer installed on the EC towers and from webcam images at the sites."

Figure 7: Please double-check that the colours used for the three sites are the same as those used in Figure 3.

Response 11: We adjusted the color scheme to be consistent with Figure 3.